# GENERATIONPROGRAMS: Fine-grained Attribution with Executable Programs

**David Wan**[1]   **Eran Hirsch**[2]   **Elias Stengel-Eskin**[1]   **Ido Dagan**[2]   **Mohit Bansal**[1]
[1]UNC Chapel Hill   [2]Bar-Ilan University

## Abstract

Recent large language models (LLMs) achieve impressive performance in source-conditioned text generation but often fail to correctly provide fine-grained attributions for their outputs, undermining verifiability and trust. Moreover, existing attribution methods do not explain how and why models leverage the provided source documents to generate their final responses, limiting interpretability. To overcome these challenges, we introduce a modular generation framework, GENERATIONPROGRAMS, inspired by recent advancements in executable "code agent" architectures. Unlike conventional generation methods that simultaneously generate outputs and attributions or rely on post-hoc attribution, GENERATIONPROGRAMS decomposes the process into two distinct stages: first, creating an executable program plan composed of modular text operations (such as paraphrasing, compression, and fusion) explicitly tailored to the query, and second, executing these operations following the program's specified instructions to produce the final response. Empirical evaluations demonstrate that GENERATIONPROGRAMS significantly improves attribution quality at both the document level and sentence level across two long-form question-answering tasks and a multi-document summarization task. We further demonstrate that GENERATIONPROGRAMS can effectively function as a post-hoc attribution method, outperforming traditional techniques in recovering accurate attributions. In addition, the interpretable programs generated by GENERATIONPROGRAMS enable localized refinement through modular-level improvements that further enhance overall attribution quality.[1]

## 1 Introduction

Recent large language models (LLMs) have demonstrated impressive performance in generation tasks. However, these models often struggle to attribute their outputs accurately through proper citations to source material, particularly in attributed generation tasks such as question answering (QA) (Bohnet et al., 2023; Gao et al., 2023b; Slobodkin et al., 2024) and summarization (Laban et al., 2024). Precise attributions, for example in the form of citations, have become increasingly important with the advent of agentic "deep research" workflows (OpenAI, 2025; Gemini, 2025), which explore the web to retrieve diverse sources and synthesize citation-supported reports. Consequently, failure to provide precise and fine-grained attributions significantly harms the faithfulness and trustworthiness of the generated content, hindering the verification of potential "hallucinations"—generated content not supported by input sources (Ji et al., 2023; Rashkin et al., 2023). Recent work suggests models achieve higher accuracy when attribution is not explicitly required (Zhang et al., 2024a), indicating a significant challenge in managing both accurate generation and faithful attribution simultaneously. For example, as illustrated in Figure 1, baseline methods often fail to provide accurate citations, resulting in incorrect or missing attributions. This problem underscores the necessity for a modular approach that decouples the generation of accurate answers from the process of producing precise and interpretable attributions.

---

[1]Our code is available at https://github.com/meetdavidwan/generationprograms.

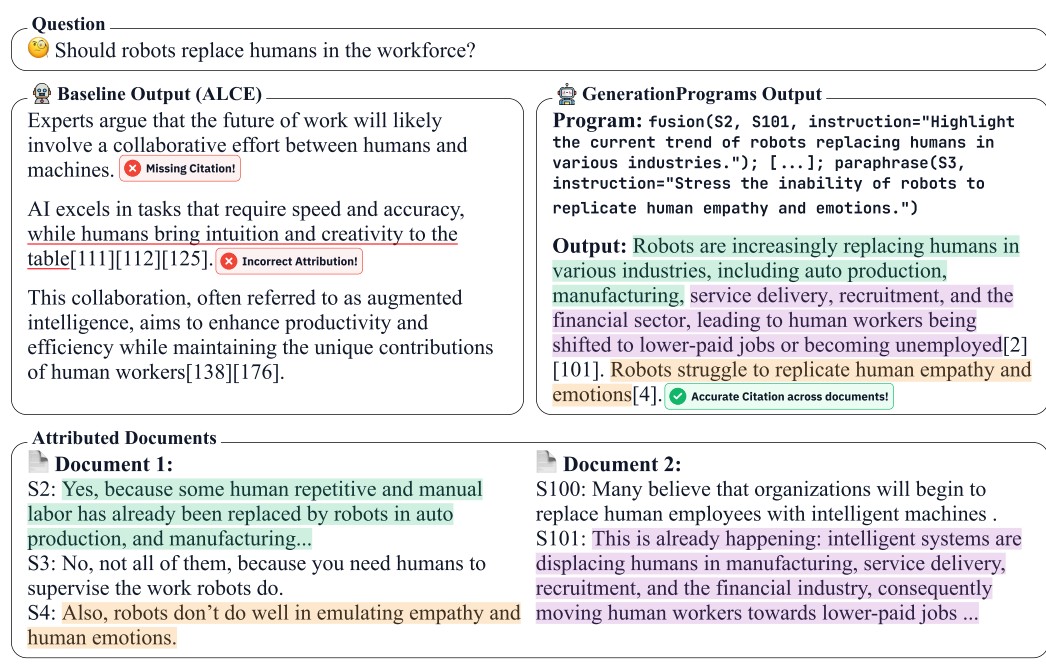

Figure 1: Example output produced by generating with citations (left) compared to GENER­ATIONPROGRAMS (right), which first generates an executable program and then performs text operations. The baseline method, ALCE (Gao et al., 2023b), may omit or generate incorrect citations (underlined), whereas GENERATIONPROGRAMS produces comprehensive and accurate citations across documents. We have color-coded the attributions in GENERA­TIONPROGRAMS for easier verification, and shortened the example to fit the page.

Furthermore, current attribution methods predominantly offer only *corroborative* attributions-identifying sources that support generated statements-rather than *contributive* attributions, which would explicitly clarify how and why specific sources led the model to its final output (Worledge et al., 2023; Cohen-Wang et al., 2024). This distinction is essential because verifying citations alone can be challenging (Rashkin et al., 2023; Liu et al., 2023a), while clearly understanding which sentences contributed allows users to directly verify the accuracy and appropriateness of generated content. Without contributive attributions, the interpretability of models remains limited, hindering fine-grained control and trust in the generation process.

Motivated by recent developments in modular "code agent" architectures—frameworks that effectively break down complex tasks into simpler executable code (Yuan et al., 2024; Yang et al., 2024; Wang et al., 2024b)—we propose GENERATIONPROGRAMS, an interpretable generation framework explicitly designed to facilitate accurate and fine-grained attribution. The text-to-code framework has led to various plan-based approaches, such as methods like "Attribute First" (Slobodkin et al., 2024) that decompose the generation process into distinct stages of content selection and then surface realization. However, these methods can be rigid in their structure, often relying on predefined templates or a fixed sequence of operations. Our work builds on the idea of explicit, executable plans for generation, drawing inspiration from the modular framework proposed in SummarizationPrograms (Saha et al., 2023). Specifically, GENERATIONPROGRAMS formulates generation as a two-stage process: first, creating a structured executable program composed of modular text operations (such as paraphrasing, compression, and fusion) explicitly tailored to a given query; and second, executing these operations through clear, query-specific instructions to produce the final response. As illustrated in Figure 1, GENERATIONPROGRAMS initially generates a program—for instance, instructing a fusion module to merge two sentences with explicit guidance on which aspects should be highlighted in the output. After execution, the resulting sentence is both instruction-consistent and faithful to the input sentences (e.g.

S2 and S101 in Figure 1), and the two sentences that were used automatically become part of the sentence-level citations (e.g. [2][101]). This structured decomposition naturally enhances attribution by explicitly tracking the source inputs used at each modular step. Consequently, attribution becomes straightforward by reviewing the explicit program trace, clearly linking each generated segment to its source sentences. The program itself also serves as an explicit explanation of the generation process, enabling fine-grained verification and facilitating targeted, localized edits.

While structured programs significantly enhance traceability and attribution quality, language models still face challenges when handling extensive input contexts. Recent advancements enable LLMs to handle extended contexts spanning millions of tokens (Gemini Team et al., 2024; OpenAI et al., 2024); however, they often struggle with reasoning over lengthy inputs, a phenomenon known as "lost-in-the-middle" (Liu et al., 2024a). To address this limitation within our structured generation approach, we investigate whether decoupling the task of identifying relevant information from the attributed generation process can further enhance attribution accuracy. Specifically, we apply summarization techniques to filter irrelevant content from retrieved source documents (Gao et al., 2023b), ensuring models concentrate solely on relevant information during generation.

Empirical evaluations across two long-form QA tasks show that GENERATIONPROGRAMS substantially improves attribution quality at both document-level and sentence-level granularities, achieving gains of 31.6% and 27%, respectively, in attribution F1 over baseline methods that directly generate citations alongside outputs. Although we observe a minor trade-off in answer correctness, we demonstrate that applying summarization techniques to source documents significantly mitigates the long-context problem, balancing attribution quality and answer accuracy. Similar observations are also made when applying GENERATIONPROGRAMS to multi-document summarization tasks. Furthermore, we showcase practical applications of GENERATIONPROGRAMS, including its effectiveness as a post-hoc attribution method. Specifically, GENERATIONPROGRAMS achieves superior performance in correctly attributing previously generated sentences compared to traditional attribution methods, with the generated program serving as an interpretable explanation of the model's reasoning process. Finally, we illustrate that the contributive nature of the generated program enables fine-grained, modular-level refinement. By correcting modular executions that are not faithful to their inputs, we achieve further improvements in attribution quality, surpassing standard attribution methods that employ reranking strategies.

## 2 GENERATIONPROGRAMS

For long-form question-answering, the input consists of a query, or question, $q$ and a set of $n$ retrieved documents $D = \{D_1, D_2, \ldots, D_n\}$. The model then uses these inputs to generate an output $o$. Our method, GENERATIONPROGRAMS, operates in two main steps: generating a program plan and executing this program using neural modules. Below, we first introduce a formal definition and then describe each step in detail.

### 2.1 Program Definition

Following Saha et al. (2023), we define a program $P = \{P_i\}_{i=1}^{k}$ as a collection of $k$ trees, each corresponding to a single sentence in the output. Each tree $P_i = (V, E)$ consists of nodes $V$ representing source or generated sentences and edges $E$ representing neural modules applied to these sentences. Leaf nodes correspond to sentences extracted directly from documents $D$, intermediate nodes represent sentences generated by neural modules, and the root node represents the final generated sentence. The overall answer combines root nodes from all trees, enabling clear tracing and attribution of each output sentence back to the modules and input sentences that generated it.

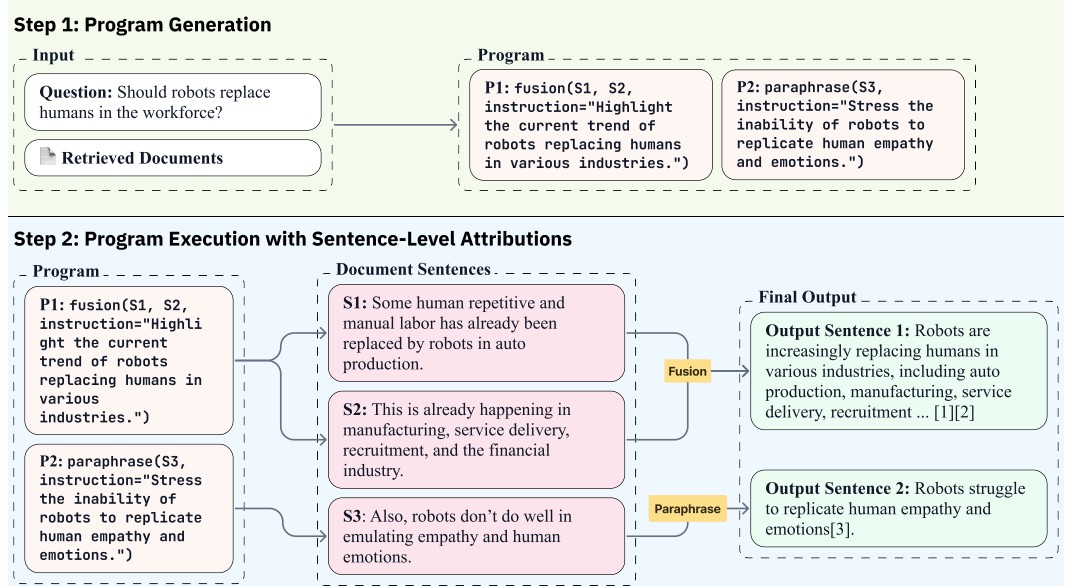

Figure 2: Illustration of GENERATIONPROGRAMS. First, an executable program is generated from the question and retrieved documents. Next, the program is executed using dedicated text-based operations to produce the final sentences. The sentences used are automatically treated as sentence-level attributions. This design promotes both interpretability and reliable attribution by tracing the program execution and citing every sentence used.

## 2.2 Program Generation

The first step of GENERATIONPROGRAMS is generating a program based on the query and retrieved documents. Specifically, we prompt a large language model (LLM) to generate Python-executable programs that correspond to individual output sentences, leveraging Python's Abstract Syntax Tree (AST) module for easy parsing. These programs may contain nested functions, allowing the output from one neural module to serve as input for another.

As illustrated in the top section of Figure 2, the generated program structure naturally results in multiple trees, each representing distinct output sentences. Additionally, our method generalizes beyond single-document summarization to query-focused and multi-document scenarios (Appendix A.1).

## 2.3 Program Execution with Neural Modules

The second step involves executing the generated program using neural modules. As depicted in Step 2 of Figure 2, GENERATIONPROGRAMS retrieves the relevant sentences and invokes corresponding modules. For example, the fusion module processes sentences in the first program, whereas the paraphrase module applies to sentences in another. Each resulting sentence attributes its content to the input sentences used.

Our neural modules build upon prior modular summarization approaches (Lebanoff et al., 2019; 2020; Jing & McKeown, 1999; 2000; Saha et al., 2023):

1. **Paraphrase:** Generates a sentence conveying same meaning with different wording.

2. **Compression:** Condenses sentences while preserving original meaning.

3. **Fusion:** Integrates multiple sentences into a single coherent sentence, resolving redundancy and reconciling differing viewpoints.

4. **Extract:** Facilitates extractive summarization to extract source sentences.

When executing neural modules, the output should be entailed by their inputs; this principle is what we refer to as *local attribution*. Ensuring local attribution guarantees that, even in the case of nested functions, each step along the program trace remains faithful with respect to its input, thereby preserving transitive attribution integrity. We analyze the local attribution performance of the modules in Appendix D and further investigate localized attribution verification and refinement strategies in Section 4.3.

## 2.4 Effectively Handling of Long-Context Using Summarization

While recent LLMs can handle extended contexts spanning millions of tokens (Gemini Team et al., 2024; OpenAI et al., 2024), these models still exhibit limitations when reasoning over lengthy inputs, a phenomenon known as "lost-in-the-middle" (Liu et al., 2024a). To mitigate this, we apply summarization techniques to source documents, reducing redundancy and filtering irrelevant information, thus also reducing the likelihood of generating unfaithful summaries (Gao et al., 2023b). Applying summarization to the retrieved documents ensures the model remains focused, potentially improving attribution accuracy.

Building upon Saha et al. (2023), we use strong query-focused summarization systems to individually summarize each source document. Document-level attribution is naturally ensured through summarization since recent methods reliably produce faithful summaries (Zhang et al., 2024b). However, precise sentence-level attribution necessitates either explicitly tracked text manipulations, as in GENERATIONPROGRAMS, or purely extractive summarization. Thus, our explicit program-based approach provides rigorous sentence-level traceability and accurate attribution.

# 3 Experimental Setup

## 3.1 Datasets

We evaluate our approach using challenging question-answering tasks involving retrieved sources. Dataset statistics are summarized in Table 4 in the Appendix. We include the ASQA dataset (Stelmakh et al., 2022), comprising factoid questions collected by Gao et al. (2023b). ASQA uses retrieved relevant Wikipedia snippets, retrieved via GTR (Ni et al., 2022), from which we select the top 10 passages for our experiments. Additionally, we use the LFQA dataset (Liu et al., 2023a) following the split curated by Slobodkin et al. (2024), which includes ground-truth documents as part of the evaluation. For summarization, we use the processed dataset by Ernst et al. (2024), a multi-document summarization (MDS) task derived from MultiNews (Fabbri et al., 2019). Additional details are in Appendix B.

## 3.2 Metrics

**Answer Accuracy.** Since both datasets include gold-standard answers, we assess the model's responses against these references using established metrics. Specifically, we use Exact Match (EM) for ASQA, and ROUGE-L (Lin, 2004) for LFQA and MultiNews MDS.

**Attribution Quality.** Following prior research (Gao et al., 2023b; Zhang et al., 2024a), we evaluate attribution quality using three model-based metrics: attribution recall, attribution precision, and attribution F1. Attribution recall checks if generated statements are fully supported by the cited sources by assessing entailment. Attribution precision ensures each citation accurately supports its respective statement. Attribution F1 is the harmonic mean of precision and recall. For ASQA, we use AutoAIS, leveraging TRUE (Honovich et al., 2022), a T5-11B model fine-tuned on various NLI datasets that correlates strongly with human judgment. Due to TRUE's input length limitations on LFQA and MDS, we instead utilize GPT-4o-based NLI evaluation, as recommended by Zhang et al. (2024a). Additionally, we measure the proportion of sentences lacking citations.

| Method | Correct | ASQA Attribution (AutoAIS) | | Correct | LFQA Attribution (GPT-4o) | | Correct | MDS Attribution (AutoAIS ) | |
|---|---|---|---|---|---|---|---|---|---|
| | EM | Attr. F1 | No Attr. ↓ | RL | Attr. F1 | No Attr. ↓ | RL | Attr. F1 | No Attr. ↓ |
| *Document-level Attributions* | | | | | | | | | |
| ALCE | **44.0** | 62.7 | 25.8 | 39.4 | 55.4 | 40.0 | **19.5** | 63.8 | 25.4 |
| ALCE + extr. summ. | 43.0 | 66.4 | 20.8 | **42.0** | 58.3 | 29.3 | 15.7 | 62.5 | 22.9 |
| GENPROG | 41.0 | 87.1 | **0.0** | 32.3 | **94.4** | **0.0** | 19.3 | **94.4** | **0.0** |
| GENPROG + extr. summ. | 39.3 | **87.2** | **0.0** | 38.4 | 87.7 | **0.0** | 17.8 | 93.7 | **0.0** |
| *Sentence-level Attributions* | | | | | | | | | |
| ALCE | **42.3** | 54.2 | 16.3 | 39.9 | 54.0 | 35.9 | **20.0** | 55.2 | 32.4 |
| ALCE + extr. summ. | 41.8 | 49.5 | 22.0 | **43.2** | 57.8 | 29.5 | 16.0 | 45.0 | 28.2 |
| GENPROG | 41.0 | **79.4** | **0.0** | 32.3 | **82.8** | **0.0** | 19.3 | 90.0 | **0.0** |
| GENPROG + extr. summ. | 39.3 | 67.2 | **0.0** | 37.9 | 65.2 | **0.0** | 17.8 | 72.7 | **0.0** |

Table 1: Results on long-form QA with document-level and sentence-level attributions.

### 3.3 Model and Methods

For our experiments, we primarily compare GENERATIONPROGRAMS against ALCE (Gao et al., 2023b), a method which directly generates answers and corresponding citations. We use GPT-4o (OpenAI, 2024) as the backbone for both methods, and the prompts are in Appendix E. We further show results with an open-source model (Llama 3.3 70B) in Appendix C.4. Both methods are provided with a one-shot example to ensure adherence to the output format. To enhance attribution quality in ALCE, we separately evaluate it at both the document and summary levels. For filtering irrelevant information with summarization, we employ extractive summarization (Erkan & Radev, 2004; Nenkova & McKeown, 2011; Cheng & Lapata, 2016; Narayan et al., 2018) on retrieved documents. In Appendix C.2, we benchmark various summarization methods for filtering retrieved content. For ASQA, we additionally perform an initial relevance filtering step using GPT-4o to exclude irrelevant documents. Detailed procedures for relevance filtering are provided in Appendix B. For replicability, we set temperature to 0 for all models.

## 4 Results

We report the attribution results in Section 4.1, and then show interesting applications of GENERATIONPROGRAMS, including post-hoc attribution (Section 4.2), and using the generated program for fine-grained module-level detection and refinement (Section 4.3). We provide additional qualitative analysis in Appendix D.

### 4.1 Attribution Results

We present the results for document-level and sentence-level attribution in Table 1. GENERATIONPROGRAMS consistently achieves significant improvements in attribution F1 compared to ALCE-based methods. Specifically, GENERATIONPROGRAMS improves document-level attribution F1 by 20.4% and 39.0% on ASQA and LFQA, respectively, and enhances sentence-level attribution by 25.2% and 28.8% on the same datasets. These results highlight GENERATIONPROGRAMS's effectiveness in improving attribution quality. However, this increased citation accuracy is accompanied by a slight reduction in answer correctness, decreasing by 2% for ASQA and 7.1% for LFQA at the document level. Nonetheless, the significant improvements in attribution quality outweigh these reductions in correctness. We further investigate the correctness evaluation in Appendix C.6, where we find that stronger LLM-based metrics and human evaluations show a smaller decrease. Furthermore, due to GENERATIONPROGRAMS's design, all generated sentences contain citations, effectively addressing a notable limitation in ALCE, where citations were omitted in 25.8% and 40% of sentences in the document-level setting for ASQA and LFQA, respectively. For MDS, the results are consistent with those observed for long-form QA tasks: we see a significant improvement in attribution quality, i.e. increasing from 63.8 to 94.4 for document-level citations.

| Attribution Method | Sentence Level | | | Output Level | | |
|---|---|---|---|---|---|---|
| | Cit. Prec. | Cit. Rec. | Cit. F1 | Cit. Prec. | Cit. Rec. | Cit. F1 |
| Semantic Similarity | 1.9 | 0.1 | 0.1 | 11.7 | 9.3 | 9.6 |
| LLM Prompt | 9.5 | 7.7 | 7.8 | 13.9 | 14.1 | 13.1 |
| GENERATIONPROGRAMS | **53.4** | **37.3** | **47.4** | **50.0** | **50.4** | **46.7** |

Table 2: Post-hoc attribution results on LFQA with oracle output.

**Effect of Summarization at Handling Long-Context.** We next examine how applying summarization to remove irrelevant content affects model performance. For ALCE, summarization enhances LFQA attribution by 2.6% at the document level and 3.3% at the sentence level, though it slightly reduces EM for ASQA by 1% at the document level and 0.5% at the sentence level. Importantly, we observe that summarization generally positively impacts attribution quality (except for sentence-level attribution on ASQA) by increasing attribution F1 and reducing the number of sentences lacking attribution. This contrasts with prior findings by Gao et al. (2023b), who reported no improvements from summarization in attribution quality. When applied to GENERATIONPROGRAMS, extractive summarization typically lowers attribution F1 slightly, but it helps mitigate the correctness trade-off caused by enhanced attribution quality. Thus, summarization provides a viable strategy for balancing answer correctness with attribution precision. For MDS, we observe that summarization does not improve the results, which agrees with prior findings that hierarchical summarization does not improve performance for summarization tasks (Wan et al., 2025).

## 4.2 GENERATIONPROGRAMS as Post-Hoc Attribution

Beyond concurrent content generation and attribution, we demonstrate that GENERATIONPROGRAMS can also be effectively adapted for post-hoc attribution, where attributions are identified after text generation (Gao et al., 2023a). Unlike traditional methods focused solely on finding supporting citations, GENERATIONPROGRAMS employs a form of *contributive attribution* (Cohen-Wang et al., 2024), explicitly identifying which sources directly contribute to the generation of specific content. This capability leverages the detailed program trace within GENERATIONPROGRAMS to precisely illustrate how input sources are integrated through text-based operations. Thus, our approach aligns with the concept of *context attribution* (Cohen-Wang et al., 2024), pinpointing exact contextual elements responsible for individual statements. This method is practically beneficial for tasks such as rapid verification and targeted refinement of generated outputs, as demonstrated in Section 4.3. Notably, our method does not require access to model logits, making it suitable for both open-source and proprietary models.

In our post-hoc attribution setup, the generated text is additionally provided as input, and GENERATIONPROGRAMS then generates a program trace to reconstruct or simulate the original output. We conduct evaluations using LFQA, which includes human annotations identifying relevant sentences for each gold answer. Given the known gold-standard sentence annotations, we measure citation precision, recall, and F1 by calculating the overlap between predicted and annotated sentence-level citations. Unlike standard attribution evaluations relying on NLI, here we specifically assess the accuracy of identifying the correct set of sentence ids. We consider two settings: (1) sentence-level evaluation, measuring citation overlap individually for each sentence, and (2) output-level evaluation, aggregating relevant sentences into sets for a more lenient assessment. We benchmark GENERATIONPROGRAMS against commonly used attribution baselines, including semantic similarity-based attribution (Reimers & Gurevych, 2019) and prompt-based attribution (Zhang et al., 2024a). For GENERATIONPROGRAMS, the attribution can be retrieved by iterating through the program trace and treating the relevant sentences as sentence-level attributions.

The results, presented in Table 2, highlight that GENERATIONPROGRAMS significantly outperforms other methods, achieving a sentence-level citation F1 of 47.4—far exceeding the next-best baseline (LLM-based prompting), which achieves only 7.8 F1 in the sentence-level setting. Additionally, since GENERATIONPROGRAMS can reconstruct sentences via its gener-

|  | ASQA | | | LFQA | | |
| Method | Correct | Attr. F1 | %Entail. | Correct | Attr. F1 | %Entail. |
| --- | --- | --- | --- | --- | --- | --- |
| ALCE | **44.0** | 54.2 | - | **39.9** | 54.0 | - |
| ALCE + refinement | 43.2 | 77.9 | - | 36.3 | 69.3 | - |
| GENERATIONPROGRAMS | 41.0 | 79.4 | 85.8 | 32.3 | 82.8 | 95.5 |
| GENPROG + module-level refinement | 41.1 | **83.4** | **87.7** | 32.4 | **85.5** | **97.2** |

Table 3: Fine-grained refinement results. In addition to attribution F1, we also show the percentages of the modules that are considered entailed for GENERATIONPROGRAMS.

ated program traces, we reconstruct the output using the generated program, and compute ROUGE scores comparing reconstructed sentences against the original gold responses. The reconstructed output results in scores of 58.9/41.2/50.7 for ROUGE-1/2/L, respectively. Regarding correctness, we observe that applying refinement actually decreases the ALCE's score. In contrast, our module-level refinement achieves a 0.1-point improvement. These results underscore GENERATIONPROGRAMS's effectiveness as a robust post-hoc attribution method, providing both accurate and interpretable contributive attributions.

### 4.3 Fine-grained Module-level Detection and Refinement

We have demonstrated that GENERATIONPROGRAMS achieves strong attribution quality for both concurrent and post-hoc attributed generations. Crucially, the accompanying program enables our method to perform *contributive* attribution—explicitly explaining why the model generates particular answers. Here, we explore an additional benefit of the program by using it to enhance attribution quality further. Since the program consists of sequences of neural module executions, maintaining faithfulness in these executions is critical for accurate attribution. Our modular approach naturally facilitates verifying the correctness of attributions and allows for proactive refinement to ensure output correctness at the module-level. We refer to this as module-level refinement.

To illustrate module-level refinement, we adopt the following experimental setup. We leverage AutoAIS to measure entailment between the generated module outputs and their respective input sentences. If a module generates a non-attributable output, we refine it using a reranking approach.[2] Specifically, we sample five candidate outputs with a temperature of 1.0 and select the first candidate considered entailed by AutoAIS. If all candidates receive identical entailment decisions, the first one is chosen by default. For comparison, we apply reranking to ALCE by sampling five entire output candidates and selecting the one achieving the highest AutoAIS score, as ALCE lacks fine-grained entailment checks at the module level. We evaluate performance using attribution precision, recall, F1 scores, and the percentage of module executions considered entailed.

The results, presented in Table 3 clearly illustrate the advantages of GENERATIONPROGRAMS. First, GENERATIONPROGRAMS significantly outperforms ALCE in attribution quality. While applying reranking to ALCE improves its attribution F1 by 23.7 and 15.3 points on ASQA and LFQA respectively, ALCE's performance still falls short compared to GENERATIONPROGRAMS without any reranking, highlighting GENERATIONPROGRAMS's intrinsic advantage. Moreover, GENERATIONPROGRAMS demonstrates notably high baseline attribution F1 and recall scores, indicating strong inherent faithfulness in module execution. Nevertheless, targeted module-level refinements further enhance performance, increasing attribution F1 by 4 points on ASQA and 2.7 points on LFQA. These improvements underscore the substantial utility of GENERATIONPROGRAMS's structured program approach, showcasing practical strategies to further enhance attribution accuracy.

An additional benefit of GENERATIONPROGRAMS for refinement is the lower computational cost. Refining modules selectively is significantly more efficient since only a small fraction requires refinement—for instance, only 4.5% of modules in LFQA need refinement.

---

[2]We do not leverage self-refinement, as prior research has indicated limited effectiveness for self-refinement (Huang et al., 2024a).

Furthermore, module refinement typically involves just a few sentences rather than entire documents, as is necessary for refining with ALCE, thus substantially reducing computational overhead.

## 5 Related work

**Attributed Generation.** Previous research on generating citations in language models primarily integrates citations into the generation process (Nakano et al., 2022; Gao et al., 2023a; Thoppilan et al., 2022; Chen et al., 2023). These methods include training specialized models (Menick et al., 2022; Zhang et al., 2024a) and employing prompting or in-context learning strategies (Gao et al., 2023b). Alternative methods focus on post-hoc attribution, adding citations after the generation process is complete (Gao et al., 2023a; Phukan et al., 2024; Qi et al., 2024). Our proposed method uniquely supports both concurrent and post-hoc attribution, explicitly designed to ensure robust attribution quality. Crucially, our approach aligns with the concept of *contributive attribution* (Cohen-Wang et al., 2024), directly identifying sources that actively influence the generated outputs of the language model. Additionally, Zhang et al. (2024c) explores the use of multiple agents to iteratively answer individual chunks of text, aiming to mitigate positional bias—a strategy similar to our approach described in Section 2.4. However, whereas their primary goal is addressing the long-context issue to enhance answer accuracy, our focus lies chiefly in improving the *interpretability of the generation* process.

**Program-based Generation.** Recent improvements in the coding capabilities of language models have sparked significant interest in program-based generation (Yuan et al., 2024; Yang et al., 2024; Wang et al., 2024b), where complex tasks are systematically decomposed and executed via structured programs. Related research also includes teaching models to utilize external tools (Schick et al., 2023; Mialon et al., 2023). In the context of text operations, Slobodkin et al. (2024); Ernst et al. (2022) similarly employ a two-step approach, involving planning and subsequent execution via a fusion module, while Saha et al. (2023) introduce a "program-then-execute" framework specifically tailored for summarization tasks. However, our proposed GENERATIONPROGRAMS method provides greater generalizability in both planning and execution phases by enabling explicit instruction-passing to neural modules. By utilizing Python-executable programs, our framework supports flexible integration of diverse modules and instructions, thus allowing more customized and adaptable output generation. Additionally, the explicit program traces generated by GENERATIONPROGRAMS enhance interpretability, allowing users to clearly understand model decisions and perform localized verification and edits as needed. We provide further explanations of the differences in Appendix A.2.

## 6 Conclusion and Future Directions

In this work, we introduce GENERATIONPROGRAMS, a framework that decomposes the attributed generation process into two distinct stages: program generation and execution via text-based neural modules. Utilizing explicit program traces, GENERATIONPROGRAMS accurately tracks document and sentence-level attributions, achieving robust attribution quality. Experimental results highlight GENERATIONPROGRAMS's superior performance in both attribution quality and citation accuracy, ensuring generated answers are reliably supported by relevant evidence. We further demonstrate that the GENERATIONPROGRAMS framework effectively handles various applications, such as providing clear post-hoc attributions that reveal the underlying rationale behind generated responses and allowing for localized verifications and edits.

As future directions, further enhancements to GENERATIONPROGRAMS could involve incorporating additional useful text-based modules, such as decontextualization (Choi et al., 2021; Gunjal & Durrett, 2024) or text simplification (Chandrasekar et al., 1996). As discussed in Section 2.2, integrating these new modules into the framework is straightforward, requiring only adjustments to the program generation prompts and implementation of new modules. While the current model focuses on sentence-level module operations, it can

easily be extended to handle phrase- or paragraph-level inputs simply by adjusting the argument granularity. Attribution benefits can be preserved as long as paragraphs maintain pointers to their constituent sentences. It is important to note, however, that tasks requiring complex reasoning, multi-hop inference, or logical puzzle-solving extend beyond pure text manipulation and are indeed outside the scope of our current framework.

Moreover, leveraging the contributive attribution via our generated program opens interesting avenues for application. For instance, as suggested by Cohen-Wang et al. (2024), one can use the generated program to reversely prune contexts to mitigate issues arising from long inputs, or for detecting adversarial modifications intended to mislead language models. In particular, it would be valuable to systematically study how best to present program-style explanations to users in a way that enhances trust and supports efficient claim verification.

## Acknowledgements

We thank the anonymous reviewers for their helpful comments. This work was supported by a Google PhD Fellowship, the Microsoft Accelerate Foundation Models Research (AFMR) grant program, NSF-CAREER Award 1846185, DARPA ECOLE Program No. HR00112390060, and NSF-AI Engage Institute DRL-2112635. The views contained in this article are those of the authors and not of the funding agency.

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

# A Additional Discussions

## A.1 Summarization, RAG, and Their Connections

We describe the tasks where applying GENERATIONPROGRAMS can improve attributions. To do so, we first introduce the formulations of summarization and retrieval-augmented generation (RAG) tasks individually, highlighting their similarities. To illustrate their connection, we intentionally overload some notation common to both tasks.

**Summarization.** Summarization typically involves taking an input document $D$ and generating an abbreviated and condensed representation, a summary $o$. Recent work (Huang et al., 2024b; Laban et al., 2024) has also focused on more challenging variants, such as multi-document summarization (Over & Yen, 2004, MDS), where the input $D$ comprises multiple documents. Compared to single-document summarization, MDS requires synthesizing content across multiple sources to handle redundant information (Fabbri et al., 2019), and highlighting contrasting or complementary viewpoints (Laban et al., 2022; Huang et al., 2024b). Additionally, summarization tasks have evolved to include query-focused or instruction controllable summarization (Zhong et al., 2021; He et al., 2022; Vig et al., 2022; Zhang et al., 2023b; Liu et al., 2024b). In these tasks, a query or instruction $q$ provides guidance for the aspects of the document(s) to be summarized.

**RAG.** Retrieval-augmented generation (RAG) (Lewis et al., 2020) is a pipeline framework for text-generation tasks, primarily used here for question-answering. Rather than solely relying on a language model's parametric knowledge, which can contain inaccuracies (Min et al., 2023; Gao et al., 2023a), RAG incorporates a retrieval component that first retrieves relevant documents $D$, and then generates answers $o$ on these retrieved sources. Both the retrieval process and the subsequent generation are conditioned on a query $q$. This is also known as the "retrieve-then-generate" pipeline.

| Dataset | Retrieved sources | Avg Source Tokens | Num Examples | Avg Num Sent |
|---------|-------------------|-------------------|--------------|--------------|
| ASQA | Passages | 1,035 | 948 | 3 |
| LFQA | Documents | 9,116 | 45 | 2 |

Table 4: Dataset statistics.

**Similarities between RAG and Summarization.**   RAG-based question answering shares structural similarities with multi-document, instruction-conditioned summarization. In RAG, retrieved documents play the role of input documents, and the guiding query $q$ helps select document segments relevant to answering the user's question. Both tasks fundamentally require extracting and synthesizing information from a set of documents, emphasizing their conceptual alignment. In fact, RAG has been applied effectively to problems like summary-of-a-haystack (Laban et al., 2024), and conversely, RAG tasks have been framed explicitly as summarization (Edge et al., 2025).

**Use of Summarization in RAG.**   In addition to the traditional summarization and RAG settings, we also explore the use of summarization in RAG as a critical post-processing step for retrieved sources (Gao et al., 2023b; Jiang et al., 2023; Xu et al., 2024; Wang et al., 2024a) to focus on filtering out irrelevant content that otherwise impedes accurate generation and citation. That is, for each document $D_i$, we perform query-focused summarization to summarize the important information regarding $q$. While previous studies (e.g., Gao et al. (2023b); Xu et al. (2024)) have primarily employed summarization to address context window constraints, we argue that summarization should also be seen as an essential extraction step, to further ease attribution.

### A.2   Comparison of GENERATIONPROGRAMS with Baseline Methods.

The most critical distinction lies in our use of optional natural language instructions within each program step, allowing the planner to communicate intent to individual modules. This design enables greater flexibility, generality, and interpretability in reasoning. We illustrate its benefits in Figure 4: although argument-based programs (e.g., sentence IDs) may indicate relevance, they often obscure the precise reasoning—especially when the same source (e.g., S3) supports multiple sentences in different ways. For instance, the first sentence may draw on the "starting points," while the second depends on length and endpoints. By incorporating explicit instructions, users can more easily verify and understand how content is generated. In comparison:

**Attribute-First (Slobodkin et al., 2024)** is limited to a single fusion operation between two sentences. In contrast, our framework supports a more general set of operations, including paraphrasing, extraction, and compression, enabling richer composition and modularity.

**Summarization Program (Saha et al., 2023)** employs multiple modules, but requires fine-tuning on synthetic program-labeled data—a process involving complex preprocessing to identify valid programs. In contrast, our method leverages the powerful zero-shot capabilities of large language models for instruction-based neural modules. Our approach enables modules to accept explicit instructions tailored to specific tasks. This flexibility empowers the planner not only to select appropriate modules but also to precisely dictate how each module should operate, facilitating the creation of more complex and context-sensitive program plans. For example, during multi-document summarization, the fusion module can be explicitly instructed to emphasize either commonalities across documents (Fabbri et al., 2019) or contrasting viewpoints (Huang et al., 2024b). Such detailed control significantly enhances the planner's capability, allowing it to construct sophisticated and highly targeted generation strategies, ultimately improving the precision, adaptability, and interpretability of the outputs.

| | Correct | ASQA Attribution (AutoAIS) | | | | Correct | LFQA Attribution (GPT-4o) | | | |
|---|---|---|---|---|---|---|---|---|---|---|
| Method | EM | Attr Prec. | Attr Rec. | Attr. F1 | No Attr. ↓ | RL | Attr Prec. | Attr Rec. | Attr. F1 | No Attr. ↓ |
| ALCE | **44.0** | 67.3 | 63.2 | 62.7 | 25.8 | 39.4 | 88.2 | 46.8 | 55.4 | 40.0 |
| GENPROG | 41.0 | 85.1 | **91.6** | 87.1 | **0.0** | 32.3 | **99.2** | **91.1** | **94.4** | **0.0** |
| abs ALCE | 43.6 | 61.7 | 64.2 | 60.7 | 20.2 | 38.8 | 84.6 | 45.1 | 52.3 | 33.0 |
| ext ALCE | 43.0 | 69.1 | 67.9 | 66.4 | 20.8 | 42.0 | 88.7 | 49.7 | 58.3 | 29.3 |
| ext-abs ALCE | 44.0 | 65.4 | 66.9 | 64.0 | 18.4 | 40.2 | 87.0 | 45.9 | 55.7 | 31.3 |
| abs GENPROG | 38.8 | 76.9 | 79.7 | 77.2 | **0.0** | 36.1 | 94.9 | 75.8 | 81.2 | 0.0 |
| ext GENPROG | 39.3 | **86.1** | 90.4 | **87.2** | **0.0** | 38.4 | 98.0 | 82.1 | 87.7 | **0.0** |
| ext-abs GENPROG | 39.6 | 82.3 | 85.7 | 82.8 | **0.0** | 39.2 | 97.8 | 83.1 | 88.1 | **0.0** |

Table 5: Attribution results on long-form QA with document-level citations.

# B  Experimental Setup Details

Here we provide additional experimental details. Sentence splitting is performed using Spacy (Honnibal et al., 2020). For GPT-4o, we use the version 2024-05-13 with temperature set to 0.

**Dataset Details.**  Dataset statistics are summarized in Table 4. We utilize the test sets for both datasets in our experiments.

**Relevance Filtering.**  For ASQA, where some retrieved documents may not be relevant, we prompt GPT-4o to assess document relevance with respect to the query. Documents deemed irrelevant are excluded from further processing.

**Summarization.**  We benchmark several summarization techniques: abstractive (Nallapati et al., 2016; Gupta & Gupta, 2019), extractive (Cheng & Lapata, 2016), and extract-then-generate (Chen & Bansal, 2018; Zhang et al., 2023a). The extract-then-generate method first applies extractive summarization, subsequently refining the output with an abstractive module.

**In-Context Examples.**  We provide the distinction of how in-context examples are used. **ALCE examples** consist of direct question-answer pairs, explicitly demonstrating the expected final output format and style. **GENERATIONPROGRAMS's examples**, in contrast, provide valid programs. The model learns to generate a process that, when executed, yields the answer.

**Ensuring Program Validity.**  We explicitly validate the generated programs by checking that all module names belong to a predefined set of supported functions and by verifying the correctness of their arguments, ensuring safe execution. The use of AST parsing is limited to converting program strings into method-name and argument tuples; all semantic validations—such as method existence and argument correctness—are handled separately.

# C  Additional Results

## C.1  Full Attribution Results.

Complete attribution results, including attribution precision, recall, and evaluations for all summarization methods, are provided in Table 5. Comparing the various summarization techniques used for post-processing retrieved sources, we observe that extractive summarization and extract-then-generate achieve the highest citation F1 scores on average. Specifically, extractive summarization improves citation F1 scores by 3.8 and 2.9 points compared to no summarization for ASQA and LFQA, respectively. This finding contrasts with previous results by Gao et al. (2023b), who reported no improvements in citation quality with summarization methods. Surprisingly, abstractive summarization does not enhance correctness.

| | ASQA | | LFQA | | Average | |
|---|---|---|---|---|---|---|
| | RL | MCS | RL | MCS | RL | MCS |
| Abstractive | **27.4** | 92.5 | **37.2** | 89.6 | **32.3** | 91.1 |
| Extractive | 24.1 | 92.0 | 32.0 | **97.8** | 28.1 | **94.9** |
| Extract-then-generate | 25.3 | **94.5** | 35.0 | 95.1 | 30.1 | 94.8 |

Table 6: Intrinsic evaluation of summaries

| | ASQA | | | | LFQA | | | |
|---|---|---|---|---|---|---|---|---|
| Method | Attr. Prec. | Attr. Rec. | Attr. F1 | %Entail. | Attr. Prec. | Attr. Rec. | Attr. F1 | %Entail. |
| ALCE | 55.3 | 59.0 | 54.2 | - | 58.2 | 55.6 | 54.0 | - |
| ALCE + reranking | 81.9 | 78.9 | 77.9 | - | 74.1 | 68.9 | 69.3 | - |
| GENERATIONPROGRAMS | 76.8 | 85.3 | 79.4 | 85.8 | 76.7 | 93.7 | 82.8 | 95.5 |
| GENPROG + module-level reranking | 80.6 | **89.3** | 83.4 | 87.7 | **80.1** | 95.2 | 85.5 | 97.2 |

Table 7: Full fine-grained refinement results. In addition to attribution F1, we also show the percentages of the modules that are considered entailed for GENERATIONPROGRAMS.

| | ASQA | | | LFQA | | |
|---|---|---|---|---|---|---|
| | Correctness | Attr. F1 | % Entail. | Correctness | Attr. F1 | % Entail. |
| Document-level Attributions | | | | | | |
| ALCE | **51.8** | 72.0 | 28.4 | 25.6 | 62.1 | 24.7 |
| ALCE + extr. summ. | 48.7 | 72.0 | 29.8 | **34.3** | 70.6 | 21.0 |
| GENPROG | 46.1 | 80.6 | **0.0** | 24.8 | **91.3** | **0.0** |
| GENPROG + extr. summ. | 45.0 | **82.7** | **0.0** | 27.2 | 89.4 | **0.0** |
| Sentence-level Attributions | | | | | | |
| ALCE | **51.0** | 56.2 | 33.1 | 32.5 | 52.4 | 58.5 |
| ALCE + extr. summ. | 47.4 | 54.3 | 38.5 | **36.3** | 57.4 | 37.1 |
| GENPROG | 46.1 | **68.9** | **0.0** | 24.8 | **86.3** | **0.0** |
| GENPROG + extractive summarization | 45.0 | 62.5 | **0.0** | 27.2 | 80.2 | **0.0** |

Table 8: Results with Llama 3.3 70B.

## C.2 Intrinsic results of summarization

To better understand the effect of summarization on retrieved sources, we conduct intrinsic evaluations of the summaries. We calculate ROUGE-L scores between generated summaries and gold-standard answers, and assess summary faithfulness relative to their source documents. Faithfulness is measured individually for each document using MiniCheck (Tang et al., 2024), a state-of-the-art method for evaluating summary faithfulness. Results are presented in Table 6. Consistent with prior findings (Zhang et al., 2023a; Xu et al., 2024), our results indicate a clear trend: extractive summarization achieves the highest average faithfulness, abstractive summarization yields the highest average ROUGE-L scores, and extract-then-generate methods offer an effective compromise by improving ROUGE-L scores relative to purely extractive methods while maintaining similarly high faithfulness.

## C.3 Full NLI Result

The complete set of NLI evaluation results, including attribution precision and recall, is presented in Table 7. In addition to our main findings in Section 4.3, we note that GENERATIONPROGRAMS exhibits significantly higher attribution recall compared to ALCE-based methods.

## C.4 Results with Open-Source Model

We report the results with an open-source model-Llama 3.3 70B-in Table 8. These findings exhibit the same trends as our primary experiments conducted with GPT-4o, described in Section 4.1. Specifically, we observe that: (1) extractive summarization effectively enhances

|  | Time(s) | Attr. F1 |
|---|---|---|
| ALCE | 3.7 | 54.0 |
| ALCE + refinement | 30.0 | 69.3 |
| GENPROG | 12.7 | 82.8 |
| GENPROG + module-level refinement | 14.5 | 85.5 |

Table 9: Latency results for refinement on LFQA.

accuracy, indicating that open-source models are similarly influenced by long-context input; (2) GENERATIONPROGRAMS significantly improves attribution quality; and (3) combining extractive summarization with GENERATIONPROGRAMS balances accuracy and attribution quality effectively. When evaluated with GPT-4o-based metrics, GENERATIONPROGRAMS demonstrates consistent performance gains, suggesting that the strong performance of GENERATIONPROGRAMS is not due to a self-evaluation bias (since here we are evaluating Llama3 outputs with a GPT4o evaluator).

## C.5 Latency

We measured the per-example runtime of both methods before and after refinement in Table 9. We found that refinement significantly increases the runtime for ALCE (approximately 8x the original runtime), as it requires generating outputs five times and performing NLI validation across all examples for reranking. In contrast, although GENERATIONPROGRAMS initially takes longer due to its two-step generation-execution approach, its structured design enables highly efficient, module-level NLI checking. Computationally-intensive operations are thus limited only to incorrect cases, with refinement adding just 1.8 seconds per example. This supports our claim that GENERATIONPROGRAMS facilitates computationally efficient refinement.

## C.6 Additional Evaluations on Correctness.

**LLM-based Correctness Evaluation.** We employ an LLM-based correctness metric introduced by Zhang et al. (2024a), shown to correlate highly with human judgments. Specifically, we compute average correctness scores for ALCE and GENERATIONPROGRAMS in the LFQA sentence-level setting, where the two methods exhibited the largest discrepancy. While Rouge-L, the standard metric, yielded scores of 39.9 for ALCE and 32.3 for GENERATIONPROGRAMS, the more nuanced LLM-based correctness metric produced much closer results—87.5 and 86.7, respectively (we found no statistically significant difference between these two correctness scores (p=0.6), indicating that the methods perform similarly in this regard). This suggests that when evaluated using a more robust and human-correlated metric, the difference between the two methods is negligible.

**Human Evaluation.** We have conducted a separate human annotation for correctness. Specifically, we asked two native English-speaking non-authors without prior knowledge of the paper to evaluate whether the generated output conveyed the same information as the reference answer. We randomly sampled 25 examples, taking both ALCE and GENERATIONPROGRAMS outputs on LFQA, where automatic metrics showed the largest gap. Following a human annotation methodology similar to that used by the authors of ALCE, annotators were asked to rate the similarity of the outputs on a scale of 1 to 5. We found that the average score for ALCE was 3.8, while GENERATIONPROGRAMS achieved an average score of 4.3 . This suggests that GENERATIONPROGRAMS actually produces statistically significantly (p=0.03) more correct answers according to this human evaluation.

The two evaluations show that GENERATIONPROGRAMS actually does not decrease correctness. Nonetheless, to maintain comparability with past work, we continue to report EM and Rouge-L in the main table.

| Dataset | Module Distribution | | | % Entailment | | |
|---|---|---|---|---|---|---|
| | fusion | compression | paraphrase | fusion | compression | paraphrase |
| LFQA | 76.8 | 16.4 | 6.8 | 96.3 | 91.7 | 93.1 |
| ASQA | 38.3 | 53.0 | 8.7 | 90.4 | 85.0 | 82.8 |

Table 10: Program statistics, including the distribution of the used modules, and the percentage of entailment of all the outputs for each module.

### C.7 Evaluation on Coherence and Fluency

Each sentence is generated independently, and we do not include an explicit contextualization step across sentences. To assess the coherence and fluency of our method despite this limitation, we utilized LLM-based evaluators on the LFQA dataset. Specifically, we employed the G-Eval (Liu et al., 2023b) prompt, which has been shown to correlate well with human judgments for evaluating text quality. We observed fluency scores of 1.3 (on a 1–3 scale) for both ALCE and GENERATIONPROGRAMS, and coherence scores of 3.0 (on a 1–5 scale) for both methods. These results suggest that, in practice, there is no significant difference in sentence-level coherence or fluency between the two approaches.

### C.8 Correlations of Attribution Metric with Human Judgment

We use LLM as judge metrics to evaluate attribution, performing binary classification to determine whether facts in the output are supported or not. To validate this metric, we collected 25 examples of attributions identified as correct by our metric and 25 identified as incorrect. These 50 examples were then randomly shuffled and presented to two native English speaker non-authors without prior knowledge of the paper. Following a setup similar to Zhang et al. (2024a), these evaluators made binary judgments as to whether the provided statements were entailed by the cited document, treating "partially supported" as "not supported". The accuracy of the metric's predictions when compared to these human-labeled instances was 78.1% (random is 50%), indicating that the metric is high-quality.

## D Additional Analyses

### D.1 Program Statistics

Program statistics are summarized in Table 10. We observe distinct distributions of module usage across the two datasets: GENERATIONPROGRAMS primarily employs the fusion module on LFQA, whereas ASQA requires compression more frequently. This highlights the adaptability of the LLM in dynamically selecting appropriate programs tailored to dataset-specific needs. Additionally, as discussed in Section 4.3, modules consistently exhibit high attribution quality. Specifically, for LFQA, each module achieves an entailment rate above 90%, while for ASQA, each module maintains an entailment rate exceeding 80%. These results underscore the robustness and reliability of the generated program structure.

**Invalid Trees.** Invalid trees are extremely rare: we find that the GPT-4o produces only one invalid program tree for ASQA and LFQA. In one case, the error stems from a missing second argument for the compression function. In the other case, the program fails to close with a final parenthesis, likely due to exceeding the maximum allowed token limit.

### D.2 Qualitative Examples

We present examples of ALCE and GENERATIONPROGRAMS outputs on the LFQA dataset in Figure 3. Additionally, we demonstrate post-hoc attribution using GENERATIONPROGRAMS in Figure 4, where we also illustrate the regeneration of the original output.

---

**Question**: Should religious education be taught in public schools?

---

**Document:**
S4: Exposing students to texts from different religions can be helpful to student learning when shared in context for the development and advancement of societies.
S5: For example, a reading from the Bhagavad Gita (Hindu text) can be used to guide a discussion on the historical time in which it was written, or an analysis of the language used.
S6: Additionally, students may be encouraged to examine the correlation between religion in school and advancement of sciences.
S7: For instance, students can research how mathematical concepts such as calculus and trigonometry came into practice with the advancement of Islam.
S8: Religious practices can be explored for historical contexts and further understanding for the development of such practices.
S12: Another way that teachers can incorporate religion in the classroom within their practice is by examining different cultures , their perspectives, and sources of motivation.
S15: When students are given the opportunity to explore diverse cultures and evaluate the causes that have led to conflicts, they are more likely to appreciate the viewpoints of others, exercise empathy and apply critical thinking skills .
S16: You can help students apply these skills when they are faced with problems that require brainstorming for a viable solution.

---

**ALCE Output:**
The question of whether religious education should be taught in public schools is complex and contentious. On one hand, exposing students to texts from different religions can enhance their understanding of historical contexts and the development of societies, as well as promote empathy and critical thinking skills[4][5][6][7][8][9][10][11][12][13][14][15][16][17][18][19][20][21][22][23].
On the other hand, the U.S. Supreme Court has consistently ruled that public schools must remain secular and cannot sponsor religious activities, as this would violate the Establishment Clause of the First Amendment[35][77][78][80][99][104][106][107][108][109][113][114][115][150][151][152]...

---

**GENERATIONPROGRAMS Program:**
```
- paraphrase(S4, instruction="Rephrase to emphasize the educational benefits of
exposing students to texts from different religions.")
- fusion(S6, S7, instruction="Combine these sentences to highlight the
historical connection between religion and the advancement of sciences.")
- compression(S12, instruction="Shorten this sentence to focus on the importance
of examining different cultures and their perspectives.")
- fusion(S15, S16, instruction="Merge these sentences to emphasize the
development of empathy and critical thinking skills through exploring diverse
cultures.")
```

---

**GENERATIONPROGRAMS Output:**
Introducing students to texts from various religions can enhance their learning by providing context for the development and progress of societies[4].
Students may be encouraged to examine the correlation between religion in school and the advancement of sciences, for instance, by researching how mathematical concepts such as calculus and trigonometry came into practice with the advancement of Islam[6][7].
Teachers can incorporate religion by examining different cultures and their perspectives[12].
When students are given the opportunity to explore diverse cultures and evaluate the causes of conflicts, they are more likely to appreciate the viewpoints of others, exercise empathy, apply critical thinking skills, and brainstorm viable solutions to problems[15][16].

Figure 3: Example on LFQA.

---

Question: where does the ohio river start and finish

---

**Document:**
S3: The Ohio River is at the boundary of the Midwestern and Southern United States, flowing southwesterly 981 miles (1582 km) long, starting at the confluence of the Allegheny & the Monongahela Rivers in Pittsburgh, Pennsylvania, and ending in Cairo, Illinois, where it flows into the Mississippi River.
S17: The Ohio River runs 981 miles ending at Cairo, IL, and is the largest tributary to the Mississippi River .

---

**Gold Response:**
The Ohio River begins at the confluence of the Allegheny and Monongahela Rivers in Pittsburgh, Pennsylvania.
It stretches for 981 miles until it ends at Cairo, Illinois where it drains into the Mississippi River.

---

**GENERATIONPROGRAMS Post-hoc Program:**
```
- compression(S3, instruction="Focus on the starting point of the Ohio River.")
- fusion(S3, S17, instruction="Combine these sentences to describe the length
and endpoint of the Ohio River.")
```

---

**GENERATIONPROGRAMS Recreated Output (Rouge-L 60):**
The Ohio River starts at the confluence of the Allegheny and Monongahela Rivers in Pittsburgh, Pennsylvania.
The Ohio River, the largest tributary to the Mississippi River, flows southwesterly for 981 miles (1582 km) from the confluence of the Allegheny and Monongahela Rivers in Pittsburgh, Pennsylvania, to Cairo, Illinois, at the boundary of the Midwestern and Southern United States.

---

Figure 4: Example of post-hoc attribution with GENERATIONPROGRAMS and its re-generated program with Rouge-L score.

| | ASQA | | LFQA | |
| --- | --- | --- | --- | --- |
| | Avg. Word | Avg. Sent | Avg. Word | Avg. Sent |
| ALCE | 31.4 | 1.7 | 87.1 | 4.7 |
| GENERATIONPROGRAMS | 41.9 | 2.3 | 140.6 | 4.8 |
| Reference | 71.8 | 3.8 | 47.5 | 2.4 |

Table 11: Average number of words and sentences.

## D.3 Stylistic Difference

We analyze potential reasons for the drop in correctness with the standard metric using GENERATIONPROGRAMS. Our core hypothesis is that the performance difference stems from a stylistic divergence between our method and ALCE, which is a direct result of their respective prompting strategies, as described in Appendix B. This distinction is critical: ALCE is optimized to mimic a specific answer style, while our method is optimized to generate a correct underlying program. The final output of our method is a result of the program's execution, which naturally leads to a different, often more verbose, stylistic presentation. This explains both the length difference and the resulting scores on style-sensitive metrics.

A result of this is that the output length of GENERATIONPROGRAMS is usually longer. We present statistical analyses of their outputs using NLTK (Bird et al., 2009) in Table 11 and observe that GENERATIONPROGRAMS tends to produce longer texts compared to ALCE.

---

Instruction: Write an accurate, engaging, and concise answer for the given question using only the provided search results (some of which might be irrelevant) and cite them properly. Use an unbiased and journalistic tone. Always cite for any factual claim with the corresponding passage number. When citing several search results, use [1][2][3]. Cite at least one document and at most three documents in each sentence. If multiple documents support the sentence, only cite a minimum sufficient subset of the documents.

*EXAMPLES**
{examples}

*YOUR TASK**
Question: {question}

Passages:
{context}

Output:

---

Figure 5: Prompt for ALCE.

## E  Prompts

We include the prompt for ALCE in Figure 5, GENERATIONPROGRAMS in Figure 6, the modules for GENERATIONPROGRAMS in Figure 7, and the summarization method in Figure 8.

You are given a document consisting of passages and a specific question. Your task is to write an accurate, engaging, and concise answer to the given question that synthesizes the information from the document. Instead of providing the final answer directly, output a list of Python function calls that, when applied to the sentences, produce the final answer. The arguments should be either sentence indices (e.g., S1) or other function calls. If you include an instruction in a function call, it must start with instruction="YOUR INSTRUCTION".

Your available functions:
1. **paraphrase(sentence, instruction=None)**
Purpose:* Rephrase the given sentence while preserving its original meaning.
Optional:* You can specify an instruction for a desired style or syntactic structure (e.g., instruction="YOUR INSTRUCTION").

2. **compression(sentence, instruction=None)**
Purpose:* Compress the given sentence to produce a shorter version that retains the essential content and syntactic structure.
Optional:* Include an instruction detailing which parts to preserve (e.g., instruction="YOUR INSTRUCTION").

3. **fusion(sentence_1, sentence_2, ... sentence_n, instruction=None)**
Purpose:* Merge multiple sentences into a single sentence. The sentences might convey similar or complementary information.
Optional:* Provide an instruction on how to combine the sentences, such as which parts to prioritize (e.g., instruction="YOUR INSTRUCTION").

*Careful:**
- **[Format]** Format your output as a bullet-point list, where each bullet point is a single sentence. For each sentence, you must output a series of Python function calls that, when executed, produce the final answer sentence. Each bullet should start with a "-" followed by the function calls without any additional content.
- **[Function Nesting]** You can nest functions as needed. The arguments for any function may be either a sentence identifier (from the document) or the output of another function call.

*EXAMPLES**
{examples}
*YOUR TASK**
Question:
{question}
Document:
{context}

Output:

Figure 6: Prompt for running GENERATIONPROGRAMS.

| | |
|---|---|
| compression | You are a sentence compression assistant. Your task is to generate a compressed version of the provided sentence while maintaining its original meaning. If an optional instruction is provided, ensure that your output adheres to it. Do not include any additional commentary or extra text. Your response should consist solely of the compressed sentence.
**Instruction**: [[INSTRUCTION]]
**Sentence**: [[SENTENCE]] |
| paraphrase | You are a paraphrasing assistant. Your task is to generate a paraphrased version of the provided sentence while strictly preserving its original meaning. If an optional instruction is included, ensure that your paraphrase adheres to it. Do not introduce any additional information or commentary. Your output should be only the paraphrased sentence, with no extra text or labels.
**Instruction**: [[INSTRUCTION]]
**Sentence**: [[SENTENCE]] |
| fusion | You are a sentence fusion assistant. Your task is to generate a single sentence that fuses the information from the provided multiple sentences (each on a new line). Merge similar information while ensuring that all differences are retained. If an optional instruction is provided, adhere to it. Your response should consist solely of the fused sentence, without any additional commentary or labels.
**Instruction**: [[INSTRUCTION]]
**Sentence(s)**: [[SENTENCE]] |

Figure 7: Prompt for modules.

| | |
|---|---|
| Extractive | Given the following document and the question "question", extract numsent sentences from the passage that can answer the question. Do not change the sentences and copy the sentences exactly as they are. You should format your output as a bullet point list, where each bullet point is a single sentence. Each bullet should start with a "-" followed by the sentence.
**Document:** {context} |
| Abstractive | Given the following document and the question "question", summarize the following document within numsent sentences that can answer the question. You should format your output as a bullet point list, where each bullet point is a single sentence. Each bullet should start with a "-" followed by the sentence.
**Document:** {context} |
| Extract-then-Generate | Given the following document, the question "question", and the extrated summary based on the document, revise the summary into numsent sentences that can answer the question. The revised summary must incorporate all the key information from the extracted summary. You should format your output as a bullet point list, where each bullet point is a single sentence. Each bullet should start with a "-" followed by the sentence.
Document: {context}
Extractive Summary: {summary} |

Figure 8: Prompt for summarization.

