# OpenReview forum: "GenerationPrograms:  Fine-grained Attribution with Executable Programs"
_colmweb.org/COLM/2025/Conference — COLM 2025_

### Official Review · Reviewer_gWx6 · 2025-05-12

**Rating:** 5
**Confidence:** 3
**Ethics Flag:** 1

**Summary:**

The paper proposes GenerationsProgram, a two-stage framework that (i) prompts an LLM to emit an executable Python “program plan” made of modular text operators (fusion, compression, paraphrase, extract) and (ii) executes that program to obtain the final answer. Because every intermediate node in the program tree retains the sentence IDs it touched, the system delivers contributive, sentence-level citations “for free.” Experiments on two long-form QA benchmarks (ASQA and LFQA) show large gains in attribution F1—+31.6 pts at the document level and +27 pts at the sentence level—while removing almost all “no-citation” cases﻿. The program trace can also be generated post-hoc to retrofit citations (47.4 F1 vs 7.8 for the best baseline)﻿, and it enables cheap, module-level reranking that adds another 2–4 F1 points at negligible compute cost﻿.

**Questions To Authors:**

1) Accuracy–attribution trade-off: Can you report answer EM/ROUGE after module-level refinement? Does reranking hurt correctness further?
2) Backbone diversity: How does the framework perform with an open-source model e.g. Llama or DeepSeek. Would also be interesting to show performnce with reasoning / frontier models e.g. DeepSeek-R1, o3, Gemini-2.5 or Claude-3.7 and with smaller models.
3) Runtime: What is the median end-to-end latency and GPU memory footprint per query versus ALCE, with and without refinement?
4) Program validity: How often does the LLM emit invalid Python or non-entailed intermediate steps, and how do you recover?
5) Security: Executing arbitrary code generated by an LLM can be risky. Did you sandbox execution or restrict the AST? What are the risks you see and how could these be mitigated?
6) Human evaluation: Have you measured whether users find the program-style explanations more trustworthy or easier to audit than plain citations? How does human evaluation relate to LLM judgement?
7) Multi-sentence operations: Can the modules handle paragraph-level fusion/compression? If not, how would you extend them?
8) Language and domain: Any preliminary results on non-English corpora or scientific/medical QA where paraphrase quality and citation stringency differ?

**Reasons To Accept:**

1)	Clear technical novelty: first work to cast attributed generation as code-agent planning with executable, instruction-passing modules, extending earlier “summarization programs” to multi-document QA and RAG﻿.
2)	Large empirical wins in document-level citation, sentence-level citation, and missing citations﻿.
3)	Versatile: works both during generation and as a post-hoc attribution tool, outperforming semantic-similarity and prompt baselines by a wide margin on LFQA﻿
4)	Interpretability & controllability: explicit program trace doubles as an explanation and allows targeted “module reranking” that boosts F1 further with extra modules refined﻿
5)	Thorough analysis: ablations on three summarization filters, intrinsic faithfulness with MiniCheck, module-usage statistics, and cost discussion showing modest overhead﻿
6)	Reproducibility: prompts and code as supplementary material and promised to be open-sourced

**Reasons To Reject:**

1)	Accuracy trade-off: answer EM/ROUGE drops by a few pts relative to ALCE﻿ and is not completely recovered by summarization.
2)	Limited generality evidence: all experiments use the proprietary GPT-4o backbone; no tests on open-source or smaller models.
3)	Dataset scope: only two English Wikipedia-based QA sets; no evaluation on summarization, dialogue, or non-English corpora.
4)	Evaluation dependence on LLMs: LFQA attribution judged with GPT-4o NLI, creating potential self-evaluation bias.
5)	Latency & engineering details: the paper claims minimal overhead but gives no wall-clock numbers or memory profile; security of executing LLM-generated Python is not addressed.
6)	Program robustness: failure modes (invalid AST, infinite loops) and their handling are not reported.
7)	Post-hoc F1 still moderate: F1 in oracle setting signals room for improvement and unclear practical utility.
8) 	Human factors absent: no user study to confirm that program traces are actually more interpretable or improve trust and how well correlated LLM judges are with human evaluations

---

> ### Author Response · Authors · 2025-06-01
> **Response to Reviewer gWx6 (Part 1)**
>
> We thank the reviewer for the helpful comments and appreciate recognizing our method’s "clear novelty" and "large empirical win."
>
> > **W1+Q1: Accuracy Tradeoff**
>
> The outputs of the two methods differ stylistically, which impacts evaluation metrics. We present statistical analyses of their outputs using NLTK in Table A and observe that GenerationProgram tends to produce longer texts compared to ALCE. A plausible explanation is that ALCE uses answer-level in-context learning (ICL), directly guiding the model to generate outputs closely matching the expected answer format. In contrast, GenerationProgram employs program-level ICL, without a strict guarantee that executing the generated program will produce outputs matching the desired final form.
> To further verify and mitigate stylistic biases, we employ an LLM-based correctness metric introduced by Zhang et al. (2024a), shown to correlate highly with human judgments. Specifically, we compute average correctness scores for ALCE and GenerationProgram in the LFQA sentence-level setting, where the two methods exhibited the largest discrepancy. While Rouge-L, the standard metric, yielded scores of 39.9 for ALCE and 32.3 for GenerationProgram, the more nuanced LLM-based correctness metric produced much closer results—87.5 and 86.7, respectively. This suggests that when evaluated using a more robust and human-correlated metric, the difference between the two methods is negligible. Nonetheless, to maintain comparability with past work, we continue to report, we continue to report EM and Rouge-L in the main table, and we will include this discussion in the final camera-ready draft.
>
> Additionally, we further analyze the accuracy tradeoff resulting from refinement by computing Rouge-L scores for LFQA after applying refinement. Specifically, the Rouge-L score for ALCE decreases from 39.9 to 36.3, while for GenerationProgram, it changes slightly from 32.3 to 32.4. The observed decrease for ALCE highlights the inherent tradeoff between accuracy and attribution, a phenomenon also noted in GenerationProgram in Lines 218–220. Importantly, applying refinement to GenerationProgram outputs does not lead to any further significant accuracy reduction.
>
>
> > **W2+Q2: Open-source models**
>
> We conducted the main experiment using the Llama 3.3 70B model, with results presented in Table A. These findings exhibit the same trends as our primary experiments conducted with GPT-4o, described in Section 4.1. Specifically, we observe that: (1) extractive summarization effectively enhances accuracy, indicating that open-source models are similarly influenced by long-context input; (2) GenerationProgram significantly improves attribution quality; and (3) combining extractive summarization with GenerationProgram balances accuracy and attribution quality effectively. We will include these findings in the final camera-ready version.
>
> | Granularity | Model | RL | Attr. F1| No Attr. ↓ |
> |-|-|-|-|-|
> | Document    | ALCE |25.6 |62.1 |24.7 |
> | Document    | ALCE + extr. summ| **34.3** |     70.6 |       21.0 |
> | Document    | GenProg|     24.8 | **91.3** |    **0.0** |
> | Document    | GenProg + extr. summ |     27.2 |     89.4 |    **0.0** |
> ||||||
> | Sentence    | ALCE                 |     32.5 |     52.4 |       58.5 |
> | Sentence    | ALCE + extr. summ    | **36.3** |     57.4 |       37.1 |
> | Sentence    | GenProg              |     24.8 | **86.3** |    **0.0** |
> | Sentence    | GenProg + extr. summ |     27.2 |     80.2 |    **0.0** |
> **Table A.** LFQA results with Llama 3.3 70B.
>
>
> > **W3+Q8: Dataset scope**
>
> We selected the two datasets because they are among the most commonly used and challenging benchmarks for question-answering tasks. We will include additional datasets, such as summarization, e.g. MultiNews similar to Attribute-First (Slobodkin et. al., 2024), before the end of the rebuttal period.
>
> > **W4: Evaluation dependence on LLMs and potential self-evaluation bias**
>
> We employed established automatic metrics that have shown high correlations with human judgments (Honovich et al., 2022; Zhang et al., 2024a; Lines 191–194). To further address potential concerns about self-evaluation bias, we conducted additional experiments during the rebuttal phase using the Llama 3.3 70B model. When evaluated with GPT-4o-based metrics (Table 1), our method demonstrates consistent performance gains, suggesting that the strong performance of GenerationPrograms is not due to a self-evaluation bias (since here we are evaluating Llama3 outputs with a GPT4o evaluator).

---

> > ### Author Response · Authors · 2025-06-01
> > **Response to Reviewer gWx6 ( Part 2)**
> >
> > > **W5+Q5: Latency**
> >
> > We measured the per-example runtime of both methods before and after refinement in Table B. We found that refinement significantly increases the runtime for ALCE (~8x), as it requires generating outputs 5 times and performing NLI across all examples for reranking. In contrast, although GenerationProgram initially takes longer due to its 2-step generation-execution approach, its structured design enables highly efficient, module-level NLI. Computationally-intensive operations are thus limited only to incorrect cases, with refinement adding just 1.8 sec per example. This supports our claim that GenerationProgram facilitates computationally efficient refinement (Lines 305–309). We will include these results in the final version.
> >
> > | | Time (s) | Attr. F1 |
> > |-|-|-|
> > | ALCE|3.7 | 54.0 |
> > | ALCE + refinement|     30.0 | 69.3 |
> > | GenProgram|12.7 | 82.8 |
> > | GenProgram + refinement |14.5 | 85.5 |
> > **Table B.** Runtime performance.
> >
> > > **W6+Q4: Analysis of invalid trees and Security and program robustness**
> >
> > We found that the model produces only 1 invalid program tree per dataset. In 1 case, the error stems from a missing second argument for the compression function. In the other case, the program fails to close with a final parenthesis, likely due to exceeding the maximum token limit.
> >
> > We explicitly validate the generated programs by checking that all module names belong to a predefined set of supported functions and by verifying the correctness of their arguments, ensuring safe execution. The use of AST parsing is limited to converting program strings into method-name and argument tuples; all semantic validations—such as method existence and argument correctness—are handled separately. We will clarify these implementation details.
> >
> > > **W7: Moderate Post-hoc F1**
> >
> > We emphasize that GenerationProgram’s performance was already notably high even before post-hoc refinement (Lines 295–298). As Table 3 indicates, both attribution F1 and the percentage of entailed information already surpass ALCE’s post-refinement performance, demonstrating superior attribution capabilities. Specifically, the percentage entailed reached 95.5% on LFQA. Nevertheless, post-hoc refinement further improves these scores by an additional 4% and 2.7% across the two datasets, respectively (Lines 301–302).
> >
> > > **Q7: Multi-sentence extension**
> >
> > While the current GenerationProgram focuses on sentence-level module operations, it can easily be extended to handle phrase- or paragraph-level inputs simply by adjusting the argument granularity. Attribution benefits can be preserved as long as paragraphs maintain pointers to their constituent sentences. We will incorporate this potential extension into our discussion of future work. Doing so will also require identifying appropriate datasets that allow for meaningful evaluation of multi-sentence or paragraph-level generation, which is currently beyond the scope of this work.
> >
> > > **W8a: Human evaluation of program-style explanations**
> >
> > We appreciate this insightful suggestion. Our primary motivation for employing the program structure was to improve attribution quality, as demonstrated by the gains in attribution F1 and reductions in “No Attribution” rates (Section 3.2). While our current work does not directly expose the underlying program to end users, we do illustrate its potential benefits in Figure 4.
> > In that example, we present both the program and the generated output. If the program were hidden from the user, the output might resemble that of any standard model, requiring the user to manually trace each claim back to the source text using span pointers. This can be labor-intensive—especially when the same source (e.g., S3) supports multiple sentences in different ways. For instance, the first sentence depends on the “starting points,” while the second involves both length and endpoints.
> > However, when the program and its instructions are made visible, the rationale behind each sentence becomes significantly easier to interpret. This structured view can help users verify content more efficiently and understand the generation process with greater transparency.
> >
> > We agree this direction holds promise. In particular, collaboration with HCI experts would be valuable to systematically study how best to present program-style explanations to users.
> >
> > > **W8b: Human evaluation of metric correlations**
> >
> > We use LLM as judge metrics to evaluate attribution (Line 185-194), performing binary classification to determine whether facts in the output are supported or not. To validate this metric, we are currently running a human evaluation wherein two non-author native speaker annotators are performing the same task, and we will evaluate the correlation between the human annotators and the LLM judgments, which (based on our own qualitative examination of the outputs) we expect to be high. We will update our response as soon as the human annotations are finished, before the end of the rebuttal period.

---

> > > ### Author Response · Authors · 2025-06-06
> > > **Follow up with additional promised experiments**
> > >
> > > \> **Additional datasets**
> > >
> > > Based on this suggestion, we have evaluated our approach on Multinews, a multi-document summarization dataset collated by Attribute-First. The results, presented below, are consistent with those observed for long-form QA tasks. Specifically, we see a significant improvement in attribution quality, i.e. 63.8 \-\> 94.4 for document-level citations.
> > >
> > > |              | rl   | attr f1 | non-attr percentage |
> > > |--------------|------|--------|----------------|
> > > | ALCE         | 19.5 |   63.8 |           25.4 |
> > > | program      | 19.3 |   94.4 |            0.0 |
> > > | ALCE sent    | 20.0 |   55.2 |           32.4 |
> > > | program sent | 19.3 |   90.0 |            0.0 |
> > >
> > > \> **Human evaluation of attribution**
> > >
> > > To further assess attribution quality, we conducted a human evaluation. We collected 25 examples of attributions identified as correct by our metric and 25 identified as incorrect. These 50 examples were then randomly shuffled and presented to two native English speaker non-authors without prior knowledge of the paper. Following a setup similar to LongCite (Zhang et al., 2024a), these evaluators made binary judgments as to whether the provided statements were entailed by the cited document, treating "partially supported" as "not supported". The accuracy of the metric's predictions when compared to these human-labeled instances was 78.1% (random is 50%), indicating that the metric is fairly high-quality. Moreover, we emphasize that AutoAIS, the metric used for sentence-level attribution, has been thoroughly tested and shown to have high correlations with human judgments (Line 192).

---

> > > > ### Author Response · Authors · 2025-06-09
> > > > **Friendly reminder to Reviewer gWx6**
> > > >
> > > > Given that only one day remains in the rebuttal period, we wanted to check in and see if our response and additional results above have addressed your comments and will allow you to revisit your score — otherwise we’re happy to continue discussing in the remaining day!

---

> > > > > ### Comment · Area_Chair_VgL1 · 2025-06-09
> > > > >
> > > > > Another friendly reminder to the reviewer to follow up on the authors' messages as soon as possible to facilitate healthy peer review. Thanks!

---

> > > > > ### Author Response · Authors · 2025-06-10
> > > > > **Last Follow-up Reminder: Discussion period ending in a few hours.**
> > > > >
> > > > > Since the rebuttal period is ending in a few hours (June 10th AoE) we are sending another reminder to see whether our additional results and our clarifications have addressed your comments and will allow you to increase your score. Given the high effort of our response (covering many points raised in your review) we hope you will have a chance to read the response.

---

### Official Review · Reviewer_TDKU · 2025-05-12

**Rating:** 6
**Confidence:** 4
**Ethics Flag:** 1

**Summary:**

The paper’s goal is to generate texts with accurate fine-grained attributions to the source documents. Given the user’s query and the documents (already retrieved before generation starts):

* Use a prompted LLM to generate k programs, each corresponding to an output sentence. The program (e.g. `fusion(S1, S2, instruction=”Highlight the trend of robots replacing humans.”)`) contains the source documents (`S1` and `S2`) and what to do with them (`fusion` = integrating them into a single coherent sentence). The sources become the attributions/citations of the sentence. The instruction gives a semantic outline of the output sentence, ensuring a narrative flow.
* (optional) Summarize long sources.
* Use a prompted LLM to execute each program separately.

The approach is evaluated on long-form QA. It improves attribution quality as measured by NLI scores against the attributed sources, though the end task accuracy drops a bit. The approach can also be used for post-hoc attribution and refinement by giving the generated text as additional input.

**Questions To Authors:**

1. The programs are defined as trees, but the examples only show flat functions with 2-3 arguments. Are there cases where a recursive tree is needed?

**Reasons To Accept:**

1. **The proposed method is effective at generating attributed sentences.** The program acts as an interpretable causal link from the source documents to the sentence, so the attribution is likely to be more accurate compared to post-hoc attribution (generate first then link to the sources).

2. **The proposed method still gives the LLM some freedom on how to structure the response.** The model is given all the sources, and it can choose any subset of the sources to base the response on, and the responses do not have to be a strict summary of the sources. The “instruction” field in each program contains the guideline of how the sentence and thus overall response should flow.

3. The method can be adapted to **take existing generated text as additional input** and attribute or refine it. Experiments on post-hoc attribution and fine-grained refinement show good improvements in terms of attribution metrics.

**Reasons To Reject:**

Overall, while the method improves attribution, **it might be sacrificing many other properties** such as the end task accuracy, coherence, ~and factuality.~

1. **End task accuracy:** The decrease in end task accuracy (Line 219) is not fully analyzed. It would be troublesome if the individual sentences are seemingly attributed in the NLI sense but the response is not good when taken as a whole. Some qualitative analysis of the errors could reveal the issues.
    * Line 220 also lists the amount of accuracy drop (2% on ASQA and 7.1% on LFQA) and these do not look “slight” to me.
    * Extractive summarization seems to help, but this also decreases attribution by quite a lot (Table 1).
    * Note that the issue could be due to the evaluation methodology (e.g. maybe how ASQA computes exact match in long answers does not play well with the proposed method), but this has to be proven by an analysis.
    * **EDIT:** Based on the author's response, when more complex evaluators (e.g. LLM judge and human) are used, the numbers are closer to the baseline. I recommend adding these results as well as the qualitative analysis about the "drop" in EM/ROUGE-L to the paper.

2. **Coherence and fluency:** While the “instruction” field dictates the response flow, each program is executed separately (see the prompt in Figure 7). This means cross-sentence coherence could be disrupted. Fluency could also be affected. For example, certain entities (e.g. “robot” in Figures 1 and 2) could be replaced with pronouns in normal writing, but they would be repeated in the proposed approach. **There is no evaluation on coherence and fluency.**
    * Granted, this could potentially be fixed with another round of prompted rewriting, or by feeding previously generated sentences during program execution. Please let me know if the method already does this.

3. ~**Factuality:** If some of the sources are out-of-context, the LLM may incorrectly interpret the source during execution and produce unfactual claims. The fact that the programs are executed separately and the summarization of long sources could worsen this. **The attribution metrics cannot detect such factuality errors** since the sentences are properly entailed in the NLI sense.~
    * ~Compare this to traditional RAG which always sees all the sources during generation. In an ideal scenario, the model could reconcile the conflicting information across sources and interpret out-of-context sources based on other sources (though admittedly most RAG models also exhibit this type of errors.)~
    * **EDIT:** The response addresses this concern -- the method can reconcile the sources during the program generation phase, since it can see all the programs. The provided example is convincing. As for the metric, factuality losses will appear in the end task metric, so Weakness 1 still applies.

---

> ### Author Response · Authors · 2025-06-01
> **Response to Reviewer TDKU (Part 1)**
>
> We thank the reviewer for the helpful comments and acknowledgment of our method’s effectiveness and flexibility.
>
>
>
> |                   | ASQA - Avg. Word | ASQA - Avg. Sent | LFQA - Avg. Word | LFQA - Avg. sent |
> |-------------------|------------------|------------------|------------------|------------------|
> | ALCE              |             31.4 |              1.7 |             87.1 |              4.7 |
> | GenerationProgram |             41.9 |              2.3 |            140.6 |              4.8 |
> | Reference         |             71.8 |              3.8 |             47.5 |              2.4 |
>
> **Table A.** Statistics of the generated output.
>
> > **W1. End-task accuracy**
>
> The outputs of the two methods differ stylistically, which impacts evaluation metrics. We present statistical analyses of their outputs using NLTK in Table A and observe that GenerationProgram tends to produce longer texts compared to ALCE. A plausible explanation is that ALCE uses answer-level in-context learning (ICL), directly guiding the model to generate outputs closely matching the expected answer format. In contrast, GenerationProgram employs program-level ICL, without a strict guarantee that executing the generated program will produce outputs matching the desired final form.
> To further verify and mitigate stylistic biases, we employ an LLM-based correctness metric introduced by Zhang et al. (2024a), shown to correlate highly with human judgments. Specifically, we compute average correctness scores for ALCE and GenerationProgram in the LFQA sentence-level setting, where the two methods exhibited the largest discrepancy. While Rouge-L, the standard metric, yielded scores of 39.9 for ALCE and 32.3 for GenerationProgram, the more nuanced LLM-based correctness metric produced much closer results—87.5 and 86.7, respectively. This suggests that when evaluated using a more robust and human-correlated metric, the difference between the two methods is negligible. Nonetheless, to maintain comparability with past work, we continue to report, we continue to report EM and Rouge-L in the main table, and we will include this discussion in the final camera-ready draft.
>
> > **W2.Coherence and fluency**
>
> We agree with the reviewer that in our current setup, each sentence is generated independently, and we do not include an explicit contextualization step across sentences.
> To assess the coherence and fluency of our method despite this limitation, we utilized LLM-based evaluators on the LFQA dataset. Specifically, we employed the G-Eval prompt [1], which has been shown to correlate well with human judgments for evaluating text quality. We observed fluency scores of 1.3 (on a 1–3 scale) for both ALCE and GenerationProgram, and coherence scores of 3.0 (on a 1–5 scale) for both methods. These results suggest that, in practice, there is no significant difference in sentence-level coherence or fluency between the two approaches.
> We thank the reviewer for the thoughtful suggestion to incorporate a contextualization step. While having a secondary prompt to rewrite the output would break the attribution chain, we are currently implementing the approach of “feeding previously generated sentences during program execution” and hope to update with results on LFQA before the end of rebuttal period.
>
> [1] G-Eval: NLG Evaluation using GPT-4 with Better Human Alignment. Yang Liu, Dan Iter, Yichong Xu, Shuohang Wang, Ruochen Xu, Chenguang Zhu

---

> > ### Author Response · Authors · 2025-06-01
> > **Response to Reviewer TDKU (Part 2)**
> >
> > > **W3. Factuality**
> >
> > We would like to clarify that we perform preprocessing by filtering irrelevant sources for ASQA (Lines 205–206) for both ALCE and GenerationProgram, thus reducing the likelihood of generating unfaithful summaries. Furthermore, similar to "traditional RAG, which always sees all sources," our planner also considers all sources and can dynamically handle conflicts. However, instead of directly generating the final output, it generates a program.
> >
> > Furthermore, as noted by the reviewer and demonstrated in prior works \[2, 3\], traditional RAG continues to face challenges when handling conflicting information. Analogous to our demonstration of GenerationProgram’s effectiveness in localized post-hoc refinement, we conjecture that our method could also improve conflict resolution by simplifying the tracing of conflicting sources and resolving discrepancies using only the essential context.
> > We will further clarify this point in the camera-ready version.
> >
> > To illustrate this, consider the following example from ASQA:
> >
> > **Question:** What is the most played song ever on Spotify?
> >
> > **Context 1:**
> > ... **On October 15, 2016**, One Dance became the most played song ever on Spotify, with over one billion streams, overtaking Lean On. Ed Sheeran's Shape of You overtook One Dance on September 21, 2017, and was the best-performing single worldwide in 2016\. ...
> >
> > **Context 2:**
> > ... The song reached one billion streams on Spotify in June 2017. It became the most streamed song on Spotify in **September 2017**, reaching 1,318,420,396 streams and overtaking One Dance. ...
> >
> > **Program:**
> > fusion(S2, S5, instruction="Combine details to mention that Ed Sheeran's 'Shape of You' became the most streamed song on Spotify in **September 2017**, overtaking Drake's 'One Dance'.")
> >
> > **Output:**
> > Ed Sheeran's Shape of You became the most streamed song on Spotify in **September 2017**, overtaking Drake's One Dance and reaching 1,318,420,396 streams on **September 21, 2017**.
> >
> > In this case, the planner correctly identifies and reconciles conflicting information: Context 1 provides historical precedence (One Dance was the top song in 2016), while Context 2 updates that fact with newer data (Shape of You surpasses it in 2017). Our framework handles this conflict similarly to other generation methods, but with the added benefit of traceability and interpretability. The structured program—along with the accompanying instruction—clearly communicates which sources are used and why, making it easier for users to verify the final output.
> >
> > We will include this clarification and example in the final version to highlight the conflict-resolution potential of GenerationProgram.
> >
> > \[2\] Retrieval-Augmented Generation with Conflicting Evidence. Han Wang, Archiki Prasad, Elias Stengel-Eskin, Mohit Bansal.
> > \[3\] PoisonedRAG: Knowledge Corruption Attacks to Retrieval-Augmented Generation of Large Language Models. Wei Zou, Runpeng Geng, Binghui Wang, Jinyuan Jia.
> >
> > > **Q1. Recursive Trees**
> >
> > We find that for ASQA, 2.4% (48/2009) of the sentence-level trees contain other modules as arguments. This likely stems from the case that we carefully select simpler trees as in-context examples to avoid unnecessary complexity. We also note that there is not a lot of recursion, as fusion allows for multiple arguments that reduces the depth of the trees (avg=2.11, min=2, max=10). We will clarify this when introducing the structure in Section 2\. Though deeper trees are rare in our experiments, we formulate our programs as trees for generality.

---

> > > ### Comment · Reviewer_TDKU · 2025-06-04
> > > **Thank you for the response. I have a few questions.**
> > >
> > > **End-task accuracy:**
> > >
> > > * "the more nuanced LLM-based correctness metric produced much closer results—87.5 and 86.7, respectively." --- it looks like the proposed method is still lagging behind ALCE, even when a more advanced evaluator is used, though I'm not sure how to interpret this score gap (is it significant)?
> > > * Qualitatively, what are the main reasons for the accuracy drop? For example, among questions that ALCE gets right but the proposed method gets wrong, what are the most common error types? It looks like not all errors can be attributed to the stylistic differences.
> > > * A human evaluation of the long answer would have been the most ideal. I noticed in the response to Reviewer gWx6 that human evaluation of the metric correlations is in progress --- this would be meaningful if the distribution of evaluated text is similar to the output of the models in the experiments. I will wait to see the results.
> > >
> > > **Factuality:**
> > >
> > > I am now convinced that the method can reconcile the evidence during the program generation phase. This addresses the issue.

---

> > > > ### Author Response · Authors · 2025-06-06
> > > > **Response to TDKU (Part 1)**
> > > >
> > > > We thank the reviewer for the quick response and engagement in the discussion period.
> > > >
> > > > > **Promised experiment of “feeding previously generated sentences during program execution”**
> > > >
> > > > We conducted the suggested experiment where previously generated sentences were fed as context during program execution. We observed that this approach did not improve coherence and fluency. Interestingly, the attribution improves. To verify these observations, we performed a paired t-test for statistical significance. We found no significant difference for coherence (p=0.37) or fluency (p=0.08). However, the improvement in attribution was statistically significant (p=0.02).
> > > >
> > > > A qualitative review of the examples offers an explanation for these results. The wording of the generated text did not change substantially with the added context, leading to a minimal effect on coherence and fluency scores. For attribution, however, we found cases where providing previously generated sentences as context helped the modules situate themselves, enabling them to perform text operations more accurately and reliably.
> > > >
> > > > ||Attr|Coherence|Fluency|
> > > > |-|-|-|-|
> > > > | GenProgram | 94.4 |3.0 | 1.3|
> > > > | GenProgram \+ context|96.2|2.8|1.2|
> > > >
> > > > > **"The more nuanced LLM-based correctness metric produced much closer results—87.5 and 86.7, respectively." \--- it looks like the proposed method is still lagging behind ALCE, even when a more advanced evaluator is used, though I'm not sure how to interpret this score gap (is it significant)?**
> > > >
> > > > We found no statistically significant difference between these two correctness scores (p=0.6), indicating that the methods perform similarly in this regard.
> > > >
> > > > > **A human evaluation of the long answer would have been the most ideal. I noticed in the response to Reviewer gWx6 that human evaluation of the metric correlations is in progress \--- this would be meaningful if the distribution of evaluated text is similar to the output of the models in the experiments. I will wait to see the results.**
> > > >
> > > > Thanks for this suggestion. We would like to clarify that the human evaluation mentioned in the response to Reviewer gWx6 primarily focused on correlations for attribution and did not assess correctness. Therefore, we have conducted a separate human annotation for correctness. Specifically, we asked two native English-speaking non-authors without prior knowledge of the paper to evaluate whether the generated output conveyed the same information as the reference answer. We randomly sampled 25 examples, taking both ALCE and GenerationProgram outputs on LFQA, where automatic metrics showed the largest gap. Following a human annotation methodology similar to that used by the authors of ALCE, annotators were asked to rate the similarity of the outputs on a scale of 1 to 5\. We found that the average score for ALCE was 3.8, while GenerationProgram achieved an average score of 4.3 (p=0.03). This suggests that GenerationProgram actually produces statistically significantly more correct answers according to this human evaluation.

---

> > > > > ### Author Response · Authors · 2025-06-06
> > > > > **Response to TDKU (Part 2)**
> > > > >
> > > > > > **Qualitatively, what are the main reasons for the accuracy drop? For example, among questions that ALCE gets right but the proposed method gets wrong, what are the most common error types? It looks like not all errors can be attributed to the stylistic differences.**
> > > > >
> > > > > Overall, we observe that the output from GenerationProgram tends to be lengthier. For instance, in the following example, both outputs received a human rating of 4, but GenerationProgram scored lower on Rouge-L:
> > > > >
> > > > > **Reference**: Reparations can be used as a form of punishment, as evidenced by the World War I reparations paid by Germany and its allies, and by advocates for restorative justice who argue that punitive tactics like reparations have a place in this method. Reparations can serve as a bridge to healing the damage to the community and victims.
> > > > >
> > > > > **ALCE:** Reparations are not inherently a form of punishment. They are broadly understood as compensation given for an abuse or injury, aimed at redressing gross and systematic violations of human rights or humanitarian law through measures such as compensation, restitution, and public acknowledgment of past violations \[1\]. While reparations can include financial compensation and other forms of redress, their primary goal is to acknowledge victims' suffering and provide measures to restore their dignity and rights, rather than to punish the perpetrators \[1\].
> > > > >
> > > > > **GenerationProgram:**  In the early 1900s, reparations were punitive measures dictated by treaties, requiring the defeated parties in conflicts, like Germany and its allies after World War I, to make payments\[1\]. Reparations are now understood as compensation and other measures provided by the state to victims of severe and systematic human rights violations or humanitarian law breaches by the responsible parties\[1\]. Reparations, if well designed, acknowledge victims' suffering, offer measures of redress, and provide both symbolic and material compensation for the violations suffered\[1\]. Reparations in restorative justice, such as paying victims or completing community service, can help heal community damage and allow offenders to make amends\[2\].
> > > > >
> > > > > In summary, we believe these three points demonstrate that automatic metrics may not always be reliable, particularly when outputs are lengthy. When evaluated with more robust metrics and through human evaluations, the performance gap narrows and, in the case of human evaluation, even favors GenerationProgram. We will clarify and include these in the final paper.

---

> > > > > > ### Comment · Reviewer_TDKU · 2025-06-06
> > > > > > **Thank you for the updated results. I'm not sure length is the only reason for the accuracy difference though.**
> > > > > >
> > > > > > > in the following example, both outputs received a human rating of 4, but GenerationProgram scored lower on Rouge-L
> > > > > >
> > > > > > I can see how ROUGE-L used in LFQA can penalize longer outputs (fundamentally it's an F1 score, so longer outputs --> lower precision). [Slobodkin24](https://aclanthology.org/2024.acl-long.182/), which is where you sourced the data from, used ROUGE-L and BertScore (their human eval didn't judge "correctness" per se). **Do you think BertScore would be affected by length?** Would it be possible to try computing the BertScores?
> > > > > >
> > > > > > And for ASQA, the metric used is Exact Match. Since each question in ASQA has multiple gold short answers (one for each interpretation of the ambiguous question), I assume you are checking for how many of these short answers appear in the model's generated text. Is this correct? **If so, wouldn't longer responses have an advantage,** since it has more words and thus chance to contain the gold short answers?

---

> > > > > > > ### Author Response · Authors · 2025-06-07
> > > > > > > **Response to Reviewer TDKU**
> > > > > > >
> > > > > > > > **Do you think BertScore would be affected by length? Would it be possible to try computing the BertScores?**
> > > > > > >
> > > > > > > As requested, we have computed the BERTScore between the reference and the model-generated output using the official BertScore’s official script and default model. We present the results below.
> > > > > > >
> > > > > > > These results provide a more nuanced view of performance. While ALCE remains strong at the sentence level, our method now demonstrates superior performance at the document level. This addresses the previously observed large gap in the ROUGE-L metric and suggests that our method's strength lies in generating holistically coherent, document-level responses.
> > > > > > >
> > > > > > > Furthermore, we wish to clarify the objective of our evaluation. Both our human evaluation and the LLM-based metric were designed to measure agreement between the reference and the generated output, not absolute "correctness." This evaluation paradigm is philosophically aligned with metrics like ROUGE and BERTScore, which also measure similarity and overlap rather than factual accuracy.
> > > > > > >
> > > > > > > |                        | Rouge-L | BertScore |
> > > > > > > |------------------------|---------|-----------|
> > > > > > > | ALCE Doc               |    39.4 |      88.7 |
> > > > > > > | GenerationProgram Doc  |    32.3 |      89.5 |
> > > > > > > | ALCE sent              |    39.9 |      90.7 |
> > > > > > > | GenerationProgram sent |    32.3 |      89.4 |
> > > > > > >
> > > > > > >
> > > > > > > > **ASQA and other reasons for the difference**
> > > > > > >
> > > > > > > We agree with the reviewer's characterization of the official ASQA evaluation script, and so the decrease in accuracy there is not explained by length. Our core hypothesis is that the performance difference stems from a stylistic divergence between our method and ALCE, which is a direct result of their respective prompting strategies.
> > > > > > >
> > > > > > > To elaborate, we direct the reviewer to the in-context learning (ICL) examples provided in Lines 119-120.
> > > > > > > - **ALCE's examples** consist of direct question-answer pairs, explicitly demonstrating the expected final output format and style.
> > > > > > > - **Our method's examples,** in contrast, provide valid programs. The model learns to generate a process that, when executed, yields the answer.
> > > > > > >
> > > > > > > This distinction is critical: ALCE is optimized to mimic a specific answer style, while our method is optimized to generate a correct underlying program. The final output of our method is a result of the program's execution, which naturally leads to a different, often more verbose, stylistic presentation. This explains both the length difference and the resulting scores on style-sensitive metrics.
> > > > > > >
> > > > > > > We hope that this addresses your remaining questions and will allow you to revisit your score – we appreciate your continued engagement and are happy to continue this discussion during the remainder of the discussion period.

---

> > > > > > > > ### Comment · Reviewer_TDKU · 2025-06-08
> > > > > > > > **Thank you for the response. I'm increasing my score, thought having a length/style-agnostic metric for ASQA would be nice.**
> > > > > > > >
> > > > > > > > Thank you for the response. I'm now inclined to believe that the drop in the metric numbers are due to the stylistic difference. I would recommend adding the length/style-agnostic metrics mentioned in these comments to the paper. The qualitative analysis on the difference in EM / ROUGE-L should also be included. And if possible, a *larger*-scale human evaluation on the output's true "correctness" (or at least agreement with the reference) would strengthen the paper.

---

> > > > > > > > > ### Author Response · Authors · 2025-06-09
> > > > > > > > >
> > > > > > > > > We are happy to hear that our explanation has clarified the performance differences, and thank you for increasing your score. We will be certain to incorporate these suggestions into the final version of our paper.

---

### Official Review · Reviewer_BPee · 2025-05-17

**Rating:** 6
**Confidence:** 3
**Ethics Flag:** 1

**Summary:**

The paper proposes GENERATIONPROGRAMS, a modular framework that decomposes text generation into a two-stage process for improved attribution in long-form QA. Unlike prior methods that interleave generation and citation or apply post-hoc heuristics, this approach first generates a structured, executable “program plan” using modules like paraphrase, fusion, and compression, then executes it to produce the output while explicitly tracking which input sentences contribute to each generated one. This results in fine-grained, contributive attribution at the sentence level. The framework supports both concurrent and post-hoc attribution, outperforms prior methods (e.g., ALCE) in document- and sentence-level attribution F1 on ASQA and LFQA datasets, and enables localized refinements by editing individual module steps. Summarization is optionally used to filter noisy context, and the framework generalizes well to both query-focused and multi-document settings.

**Reasons To Accept:**

The core idea of explicitly structuring generation as a sequence of interpretable operations is both novel and practical. By framing output construction as an executable program of text-editing modules, the framework makes attribution *transparent and verifiable*, allowing users to trace back how and why each part of the answer was produced. This goes beyond current “supporting sentence” approaches and enables true contributive attribution.

GENERATIONPROGRAMS also has strong empirical improvements across two challenging benchmarks. It shows consistent gains in both sentence- and document-level attribution F1 compared to strong baselines. It’s also flexible: it can be used for both real-time generation and post-hoc attribution, and the modular structure allows for localized error correction or refinement.

Another strength is its compatibility with LLMs without the need for retraining. The framework uses prompting to generate Python-style programs and runs LLM-based modules zero-shot with tailored instructions. This keeps the method lightweight and easily deployable.

**Reasons To Reject:**

The most significant limitation of this paper is that it evaluates the proposed framework exclusively using GPT-4, without testing on other models to demonstrate generality. Given that the method is presented as a modular and model-agnostic framework, it is crucial to show that it works across a range of LLMs, including smaller or open-source ones like Vicuna or LLaMA. In contrast, prior benchmarks like ALCE evaluate multiple models and show attribution F1 exceeding 80 under optimized settings—much higher than the 60s cited in this paper, which appears to be based on stricter contributive attribution but is not clearly clarified. Without broader evaluation, it’s unclear whether the observed gains stem from the framework itself or simply from GPT-4’s capabilities.

Additionally, while the framework emphasizes generality across generation tasks, the modules used (e.g., fusion, paraphrase, extract) are primarily summarization-focused. This limits its applicability to tasks involving more complex reasoning or knowledge manipulation, where different types of operations may be required.

Finally, the paper does not compare against other structured or program-based generation pipelines beyond ALCE. Recent works like Slobodkin et al. (2024), Saha et al. (2023) (mentioned in the paper), or [1] propose related approaches (e.g., plan-then-generate), and a more direct comparison or discussion would help position this work within the growing space of controllable and interpretable generation frameworks.

[1] Ji et al., Towards Verifiable Text Generation with Generative Agent. AAAI 2025.

---

> ### Author Response · Authors · 2025-06-01
> **Response to Reviewer BPee**
>
> We thank the reviewer for the helpful comments and appreciate recognizing our method as "novel and practical" with "strong empirical improvements."
>
> | Granularity | Model | RL | Attr. F1| No Attr. ↓ |
> |-|-|-|-|-|
> | Document    | ALCE |25.6 |62.1 |24.7 |
> | Document    | ALCE + extr. summ| **34.3** |     70.6 |       21.0 |
> | Document    | GenProg|     24.8 | **91.3** |    **0.0** |
> | Document    | GenProg + extr. summ |     27.2 |     89.4 |    **0.0** |
> ||||||
> | Sentence    | ALCE |     32.5 |     52.4 |       58.5 |
> | Sentence    | ALCE + extr. summ    | **36.3** |     57.4 |       37.1 |
> | Sentence    | GenProg              |     24.8 | **86.3** |    **0.0** |
> | Sentence    | GenProg + extr. summ |     27.2 |     80.2 |    **0.0** |
> **Table A.** LFQA results with Llama 3.3 70B.
>
> > **Running on open-source models**
>
> We conducted the main experiment using the Llama 3.3 70B model, with results presented in Table A. These findings exhibit the same trends as our primary experiments conducted with GPT-4o, described in Section 4.1. Specifically, we observe that: (1) extractive summarization effectively enhances accuracy, indicating that open-source models are similarly influenced by long-context input; (2) GenerationProgram significantly improves attribution quality; and (3) combining extractive summarization with GenerationProgram balances accuracy and attribution quality effectively. We will include these findings in the final camera-ready version.
>
> > **ALCE showing attribution F1 exceeding 80**
>
> We note that our setting employs 10 passages (Line 178). In the ALCE paper, the best-performing model used only 5 passages. Specifically, the original ALCE results for the *Summ (10-psg)* setting—which closely aligns with our ALCE + extractive summarization configuration—show attribution precision and recall of 68.9 and 61.8, respectively. These figures are consistent with our reported citation F1 score of 66.4 in Table 1.
>
> > **The modules are primarily summarization-focused, limiting applicability to tasks involving more complex reasoning or knowledge manipulation**
>
> We agree that our modules, though initially inspired by summarization tasks (Line 133), are fundamentally designed for general text manipulation tasks. Our evaluation explicitly includes challenging long-form question-answering tasks (Section 3.1), demonstrating their applicability beyond summarization. Moreover, as indicated in our future work discussion (Lines 347–355), we plan to extend our module set to include additional text manipulation functionalities, such as decontextualization and text simplification.
> We acknowledge that tasks requiring complex reasoning, multi-hop inference, or logical puzzle-solving extend beyond pure text manipulation and are indeed outside the scope of our current framework. We appreciate this important distinction and will clarify the intended scope explicitly in our paper.
>
> > **Comparison against other generation pipelines**
>
> We would like to clearly discuss the differences between GenerationProgram and other works. The most critical distinction lies in our use of **optional natural language instructions** within each program step (Lines 146–157), allowing the planner to communicate intent to individual modules. This design enables greater **flexibility**, **generality**, and **interpretability** in reasoning. We illustrate its benefits in Figure 4: although argument-based programs (e.g., sentence IDs) may indicate relevance, they often obscure the precise reasoning—especially when the same source (e.g., S3) supports multiple sentences in different ways. For instance, the first sentence may draw on the “starting points,” while the second depends on length and endpoints. By incorporating explicit instructions, users can more easily verify and understand how content is generated.
>
> In comparison:
>
> - Attribute-First (Slobodkin et al., 2024\) is limited to a single fusion operation between two sentences (Line 327). In contrast, our framework supports a **more general set of operations**, including paraphrasing, extraction, and compression, enabling richer composition and modularity.
>
> - Summarization Program (Saha et al., 2023\) does employ multiple modules, but requires **fine-tuning on synthetic program-labeled data**—a process involving complex preprocessing to identify valid programs (Lines 329–335). In contrast, our method is **training-free**, relying solely on **in-context learning** (Lines 121–126). This aligns with the reviewer’s observation about the benefits of using LLMs “without the need for retraining,” making GenerationProgram a **lightweight and easily deployable** solution. Moreover, while Summarization Program focuses solely on summarization tasks, we demonstrate that GenerationProgram generalizes well to **long-form question answering**, highlighting its broader applicability.
>
> We will incorporate this comparative discussion in the final camera-ready version to more clearly delineate our contributions.

---

> > ### Author Response · Authors · 2025-06-06
> > **Reminder**
> >
> > As the discussion period is now underway, we would like to send a friendly reminder regarding our response, to see if we have adequately addressed the comments in the review. We hope that our responses will allow you to revisit your score.

---

> > > ### Author Response · Authors · 2025-06-09
> > > **Friendly reminder for Reviewer BPee**
> > >
> > > Given that there is only one day left in the rebuttal period (ending June 10th), we wanted to check again whether our additional experiments and responses (including **adding open-source models**) have addressed your comments. If they have, we would appreciate it if you could revisit your score accordingly -- otherwise we are happy to continue discussing in the remaining day of the rebuttal period.

---

> > > > ### Comment · Area_Chair_VgL1 · 2025-06-09
> > > >
> > > > Another friendly reminder to the reviewer to follow up on the authors' messages as soon as possible to facilitate healthy peer review. Thanks!

---

> > ### Comment · Reviewer_BPee · 2025-06-09
> >
> > Thanks for the detailed response! Most of my concerns are fully addressed. I will adjust my score accordingly.

---

> > > ### Author Response · Authors · 2025-06-10
> > >
> > > Thank you for raising your score -- we appreciate the engagement and effort on our paper and will incorporate these updates in our final write-up.

---

### Decision · Program_Chairs · 2025-07-08

**Decision:**

Accept

**Comment:**

In this paper the authors propose a framework for sentence-level answer attribution in long-form QA, which works by generating an executing a human-interpretable “program plan” made up of sub-functions such as “paraphrase,” “fusion”, and “compression.” They evaluate the approach on GPT-4 and Llama 3.3 70B (during discussion period) for sentence- and document-level attribution on the ASQA and LFQA datasets, demonstrating better performance than prior work (in particular, ALCE).

Reviewers agreed that the proposed approach addresses a practical problem in a new and interesting way, and that the approach seems to work well. Reviewers also highlighted that the approach has additional desirable qualities, such as not requiring LLM re-training, and flexibility to work for both post-hoc and concurrent attribution. Reviewers raised concerns around the comprehensiveness of the evaluation with respect to model families and comparisons to baselines, though this was satisfactorily addressed when the authors provided additional results and explanation for why some existing approaches were in fact not suitable baselines. Reviewers also raised concerns that while the approach seems to work quite well in providing useful explanations, this comes at the cost of some prediction accuracy on the question answering task itself. However, the authors provided a reasonable and well-supported explanation for this difference during the discussion phase, which I and the authors who engaged found satisfactory, especially given the novelty of the approach. The authors should be sure to include that explanation, as well as the additional results on Llama models provided during the discussion phase, in the final version of the paper if the paper is accepted.

[Automatically added comment] At least one review was discounted during the decision process due to quality]